# UniCode: Augmenting Evaluation for Code Reasoning

Xinyue Zheng [* 1]  Haowei Lin [* 1]  Shaofei Cai [1]  Yaodong Yang [1]  Zilong Zheng [† 2]  Yitao Liang [† 1]

## Abstract

Current coding benchmarks often overstate Large Language Model (LLM) capabilities due to static paradigms and data contamination, allowing models to exploit statistical shortcuts rather than genuine reasoning. To address this, we introduce UniCode, a generative evaluation framework that systematically probes LLM reasoning boundaries via: (1) multi-dimensional augmentation operators to create diverse algorithmic variants; (2) a scalable test generation pipeline achieving 94.5% correctness without human-written solutions; and (3) fine-grained diagnostic metrics for rich error signals. Our evaluation of state-of-the-art models reveals a significant 31.2% performance collapse. Critically, we observe a high variance across different reasoning axes, revealing a profound fragility under structural shifts despite surface-level robustness. Furthermore, we identify a "seed-problem regression," where models fail by defaulting to memorized seed logic and inefficient complexities. Our evaluation code is publicly available at https://github.com/grandsmile/UniCode.

## 1. Introduction

Developing intelligent systems capable of multi-step reasoning remains a cornerstone of AI research (Wei et al., 2022; Guo et al., 2025; Jaech et al., 2024; Comanici et al., 2025; Li et al., 2022c). Competitive programming has emerged as an ideal testbed for evaluating such capabilities (Li et al., 2022b; El-Kishky et al., 2025), not merely for its rigorous evaluation signals, but because it positions coding as a formal, executable interface for general problem-solving. In this context, code becomes a universal medium to ground

reasoning and formalize multi-task solutions. (Zhu et al., 2025; Quan et al., 2025).

However, current coding benchmarks suffer from an "evaluation paradox": while LLMs have nearly achieved saturation on standard coding benchmarks (Chen et al., 2021; Austin et al., 2021; Hendrycks et al., 2021), they frequently stumble during real-world interactions (Sergeyuk et al., 2025; Weisz et al., 2025). We attribute this discrepancy to three critical limitations in existing evaluation protocols: 1) data contamination and fixed design patterns, which allow models to exploit statistical shortcuts (Figure 1d); 2) limited scalability due to the high cost of human curation (Jain et al., 2024); and 3) a reliance on static datasets that fail to capture the complex algorithmic reasoning required in evolving scenarios (Fodor, 2025; de Vladar, 2016).

To address these issues, recent research has explored dataset augmentation via perturbation (Li et al., 2024a; Mirzadeh et al., 2024; Orvalho & Kwiatkowska, 2025). However, these approaches predominantly focus on surface-level variations, such as variable renaming or background rephrasing, that leave the underlying logic unchanged. Consequently, they fail to assess whether a model has truly mastered algorithmic concepts or is merely recalling specific problem structures. This necessitates a systematic framework capable of inducing deep structural transformations to rigorously probe the boundaries of model reasoning.

In this work, we introduce **UniCode**, a framework for the *Augmented Evaluation* of code reasoning (Figure 1a), which employs a generative approach to systematically stress-test LLMs under meaningful structural, compositional, and conceptual shifts. We make the following contributions:

**Systematic Task Augmentation.** We propose an augmentation methodology that transforms seed problems into a diverse array of tasks designed to expose the inherent reasoning vulnerabilities of LLMs (Section 2). Moving beyond shallow perturbations, our approach leverages evolutionary operators to restructure reasoning graph topologies. Specifically, we apply *Atomic variations* to modify task facets (e.g., narrative, rules or input scale) to test structural adaptation; *Compositional variations* to integrate multiple knowledge points, forcing models to exhibit genuine combinatorial generalization (Figure 1b). By applying these functionally meaningful transformations, UniCode systematically maps

---

[*]Equal contribution ; [†]Corresponding authors. [1]Institute for Artificial Intelligence, Peking University [2]Beijing Institute for General Artificial Intelligence. Correspondence to: Xinyue Zheng <xyzheng25@stu.pku.edu.cn>, Zilong Zheng <zlzheng@bigai.ai>, Yitao Liang <yitaol@pku.edu.cn>.

*Proceedings of the 43rd International Conference on Machine Learning*, Seoul, South Korea. PMLR 306, 2026. Copyright 2026 by the author(s).

the reasoning boundaries of models.

**Scalable and Robust Evaluation.** To overcome the bottleneck of human-curated benchmarking, we develop a stress-driven synthesis framework for autonomous test generation (Section 3). By integrating brute-force stress-filtering with multi-model consensus, UniCode achieves a 94.5% correctness rate at a marginal cost of $0.041 per problem. This framework facilitates the continuous expansion of a contamination-resistant evaluation space, maintaining its challenge as LLMs advance. The reliability of our system is grounded in expert manual validation (App. B.4) and further supported by statistical error-bound proofs (App. B.5).

**Fine-grained Diagnostic Metrics.** Current coding evaluations often rely on binary pass rates, which obscure specific model deficiencies. Instead of merely recording success or failure, our framework provides a comprehensive diagnostic toolkit that categorizes failures into modeling errors, complexity misjudgments, logic bugs, and implementation bugs (Section 5). By decomposing error types, we can effectively decouple intrinsic reasoning or implementation failures from memorization-induced biases; this uncovers the "seed-problem regression" phenomenon that remains invisible to traditional static benchmarks.

Our comprehensive evaluation of 19 LLMs yields several insights in code reasoning. First, we identify a critical vulnerability: model performance collapses when the underlying reasoning graph topology is altered (Figure 1e). Second, high performance variance (up to 61%) across different reasoning axes reveals that single-score benchmarks fail to capture the nuanced landscape of code intelligence (Figure 3). Crucially, our diagnostics reveal that failures are not random; they often manifest as "seed-problem regression," where models revert to memorized seed logic when faced with novel algorithmic structures (Figure 5). As task complexity scales, we observe a transition from isolated errors to cascading failure chains, suggesting a systemic breakdown in the models' reasoning processes (Section 5). These findings position UniCode as a vital benchmark for advancing the robustness of next-generation code agents.

## 2. Augmentation Axes for Code Reasoning

Data-driven LLMs often rely on statistical correlations over logical reasoning, faltering in complex scenarios (McCoy et al., 2019). While perturbations are standard for testing, existing benchmarks struggle to challenge increasingly robust models. To address this, we design *Atomic Augmentations* (narrative, rule, efficiency) and *Compositional Augmentations* (sequential, concept), which are crucial as they decouple memorized patterns from genuine logic, ensuring evaluation reflects reasoning rather than statistical shortcuts. Augmentation examples are illustrated in Figure 1b.

**Narrative Perturbation** This axis modifies variable names, thematic backgrounds, or injects irrelevant contextual noise without altering the underlying logic. It specifically probes whether a model suffers from "token bias" (Jiang et al., 2024) or exhibits content-agnostic reasoning. For example, we reframe the abstract *Longest Increasing Subsequence (LIS)* problem into a real-world scenario like *Identifying the Longest Growth Period in Stock Trends* (see Figure 1b). If a model fails due to this narrative shift, it indicates a reliance on near-neighbor matching (Li et al., 2024b) of familiar problem descriptions rather than a robust understanding of the logical core.

**Rule Modification** Standard programming problems often have "canonical" solutions that LLMs easily memorize. By subtly altering operational rules or boundary conditions, we invalidate these memorized paths (Dziri et al., 2023). A representative transformation is shifting *LIS* to the *Longest Non-Decreasing Subsequence*. While seemingly minor, this change shifts the comparison operator (from $>$ to $\geq$), requiring the model to re-calibrate its logical flow according to novel instructions rather than retrieving pre-trained code snippets, effectively distinguishing retrieval from reasoning.

**Efficiency Scaling** Genuine reasoning entails an awareness of the computational budget and the ability to adapt as data scale increases (Zubić et al., 2025). This axis tests if a model can transition from a naive approach to a more optimized algorithm when complexity demands it. For instance, when the input size $n$ for *LIS* scales to $10^5$, the standard $O(n^2)$ approach becomes computationally prohibitive. The model must recognize this bottleneck and pivot to a greedy strategy ($O(n \log n)$). This transition probes the model's capacity for high-level strategy selection and its understanding of algorithmic efficiency beyond simple template filling.

**Sequential Composition** This dimension involves chaining multiple distinct algorithmic steps to examine the stability of the reasoning chain. As the sequence of operations grows, the probability of failure increases—a phenomenon known as error propagation (Schaeffer et al., 2023). In a composite variant like the *Longest Bitonic Subsequence*, the model must compute the *LIS* from both the prefix and suffix and then integrate the results. Such tasks reveal the fragility of the reasoning process, as minor logical flaws that might be hidden in single-step tasks are amplified during intermediate state transfers.

**Concept Fusion** Real-world challenges often lie at the intersection of disparate domains. The fusion variant merges distinct algorithmic concepts into a single problem, creating novel combination patterns that human designers often overlook. For instance, while dynamic programming is frequently paired with string manipulation in standard datasets,

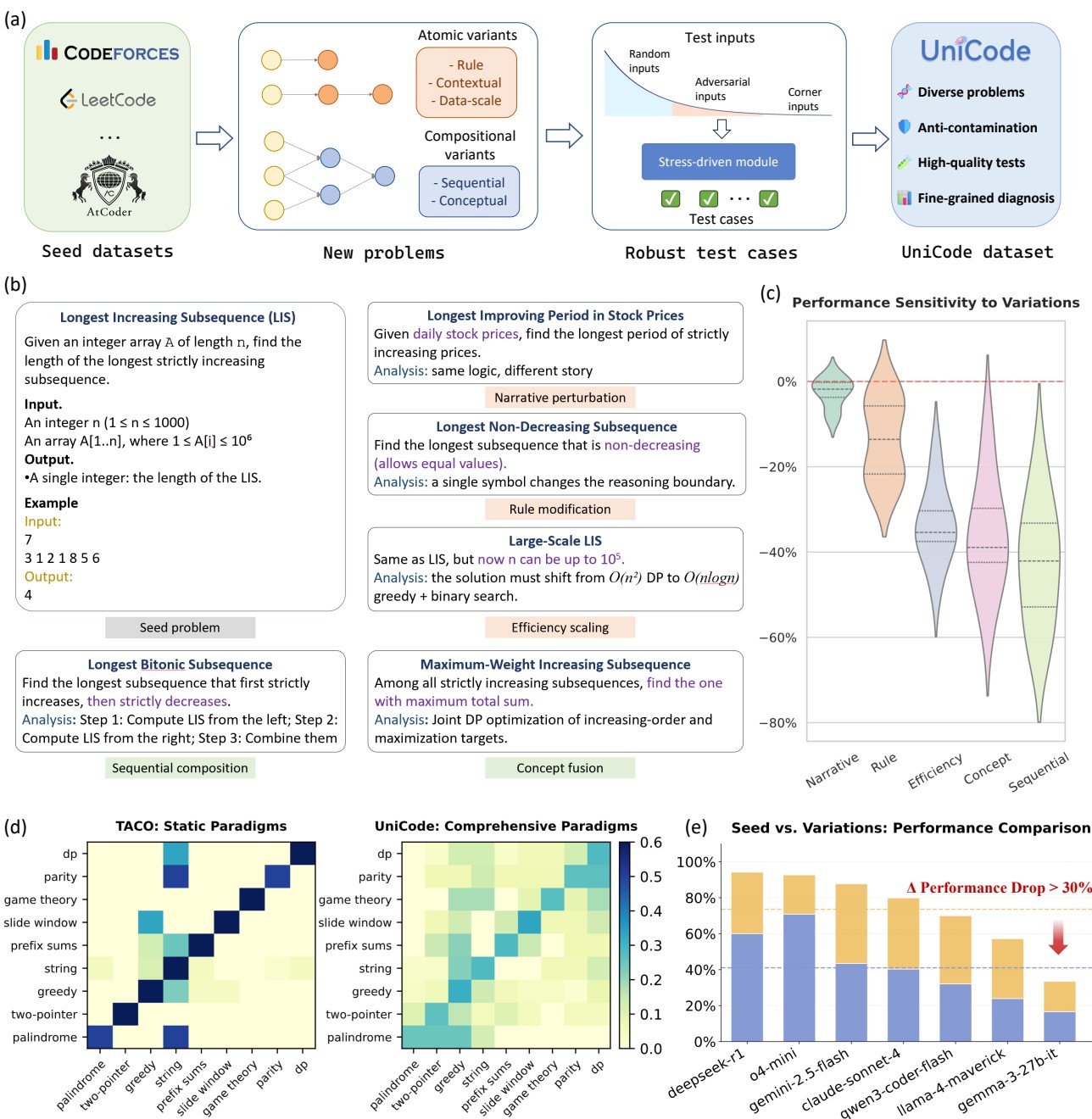

*Figure 1.* An Overview of UniCode. (a) **UniCode Framework.** We propose a generative pipeline to create a vast evaluation space via multi-dimensional augmentations to probe reasoning axes, a test generation module for scalability, and fine-grained diagnostics that expose specific failure modes for a transparent assessment of code reasoning. (b) **An Example: Probing Code Reasoning Ability via Five Augmentation Axes.** We design *Atomic Augmentations* (narrative, rule, and efficiency) to alter task facets, ranging from surface-level shifts to reconfigurations of the reasoning graph topology and *Compositional Augmentations* (sequential composition and concept fusion) to disrupt fixed algorithmic patterns and test combinatorial generalization. (c) **Model Fragility: Vulnerability to Logic Alteration.** While LLMs are robust under surface-level narrative shifts, performance drops significantly when core logic is modified. The most severe failures in sequential and fusion tasks indicate struggles with long causal chains and algorithmic integration. (d) **Beyond Human Curation: Identifying Reasoning Blind Spots.** Unlike static benchmarks (e.g., TACO (Li et al.)) with predictable patterns (e.g., *String problems frequently paired with DP but rarely with Game Theory*), UniCode explores a broader spectrum of algorithmic combinations to expose "reasoning blind spots" that human designers typically fail to cover. (e) **Performance Collapse: Memorization vs. Adaptation.** Large-scale testing reveals an average drop of $> 30\%$ when moving from seed problems to their augmented variations across LLMs. We observe a "seed-problem regression" where models default to the memorized logic of seed problems rather than adapting to new, augmented constraints. Best viewed when zoom in.

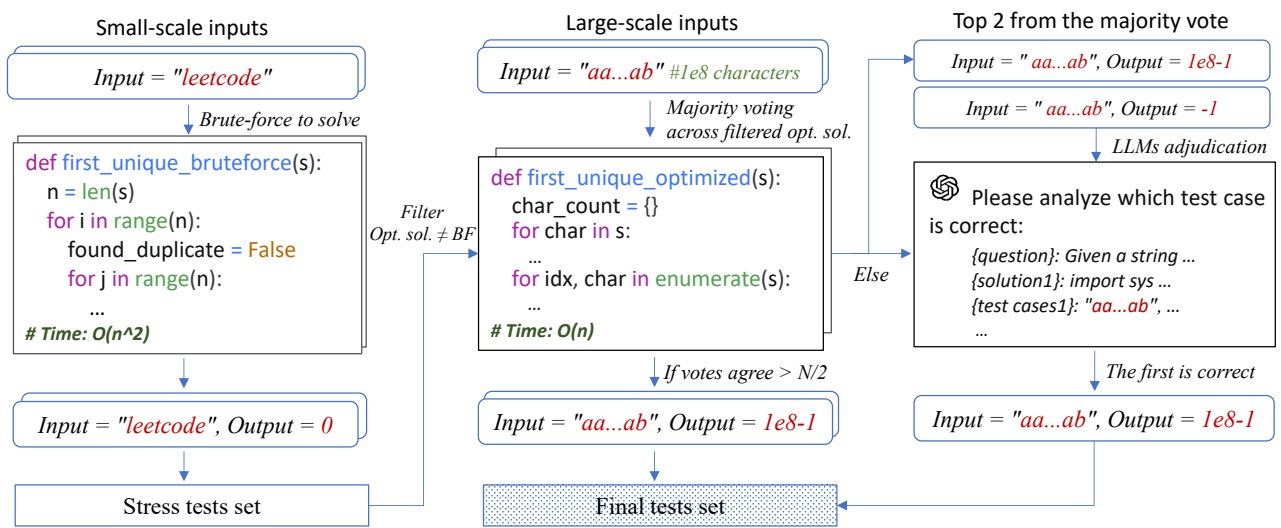

*Figure 2.* Stress-driven pipeline for ground-truth generation. *(1) Stress Testing:* we use a brute-force solver to produce trusted outputs on small inputs, which serve as stress tests to filter optimized solvers. *(2) Consensus Validation:* Remaining solvers are executed on large inputs, and the output is selected by strict majority vote. *(3) LLM Adjudication:* A powerful LLM adjudicates between conflicting outputs for inputs where no majority is reached. The effectiveness of each stage is validated by ablation (App. B.1) on human-curated datasets.

it is rarely combined with game theory or greedy algorithms, making such intersections particularly difficult to navigate. By creating these original pairings, we probe whether the model can genuinely integrate separate concepts to achieve combinatorial generalization, which is recognized as a cornerstone of human-like reasoning (Battaglia et al., 2018).

## 3. Scalable and Rigorous Test Generation

To scale evaluation effectively, an automated test generation pipeline is essential. However, the core challenge is ensuring test quality for novel problems lacking human-authored solutions. We address this by using three distinct input types to cover boundary conditions and attacks, then establishing trusted outputs through a stress-driven pipeline.

### 3.1. Input Generation

Low-quality test cases often lead to model mis-ranking (Jain et al., 2024; Liu et al., 2025; Wang et al., 2025). To ensure robustness, we construct test cases by prompting LLMs to generate inputs from three complementary sources:

- **Random Generation** ($G_{rand}$)**:** targets general correctness that samples broadly from the valid input space.

- **Adversarial Generation** ($G_{adv}$)**:** targets algorithmic inefficiencies or edge-case failures, such as boundary-value, extreme sequence lengths, and oscillating patterns.

- **Corner Generation** ($G_{corn}$)**:** targets subtle failure modes using challenging and small-scale inputs.

All candidate inputs from each source $\tau \in \{$rand, adv, corn$\}$

(input prompts detailed in App. B.7) are filtered by a verifier $V$. Let $G_\tau$ represent the initial candidate pool; the final set of verified inputs $I_\tau$ is defined as:

$$I_\tau = \{\, x \in G_\tau \mid V(x) = \text{true} \,\}.$$

To balance coverage and efficiency (Liu et al., 2023a), we specifically assemble a test suite $S$ of diverse cases: 20 random, 20 adversarial, and 10 corner cases (empirically tuned; see App. B.2). Table 3 demonstrates that this configuration ensures both correctness and coverage.

### 3.2. Ground-Truth Construction

Establishing ground-truth outputs for novel problems is challenging. We devise a multi-stage pipeline (Figure 2) that mirrors a rigorous human validation process.

**Stage 1: Brute-Force & Solver Filtration** We generate a brute-force solver $B$ via LLM to create ground-truth pairs for small-scale inputs ($I_s$). Multiple candidates and consensus ensure $B$'s reliability. We then prompt LLMs for $M$ optimized candidate solutions $\{C_1, \ldots, C_M\}$. A candidate enters the trusted pool $\mathcal{P}$ only if it matches $B$ on all $I_s$:

$$\mathcal{P} = \{C_j \mid C_j(i) = B(i)\ \forall i \in I_s\}.$$

Ablations (App. B.1) confirm this stage effectively filters correlated failures (shared flaws across optimized solvers), providing a rigorous correctness guarantee for the pipeline.

**Stage 2: Consensus on Large-Scale Inputs** For large-scale inputs ($I_\ell$) where brute-force is infeasible, we use the pool $\mathcal{P}$. The ground truth for $i \in I_\ell$ is determined by a strict majority vote ($> \lfloor N/2 \rfloor$) among the $N$ optimized solvers.

| Model | Difficulty (Pass@1) | | | Test Impact ($\Delta \downarrow$) | | | Avg. Pass@1 | Cost / Prob. ($) |
|---|---|---|---|---|---|---|---|---|
| | **Easy** | **Medium** | **Hard** | $\Delta_{Rand}$ | $\Delta_{Adv}$ | $\Delta_{Corn}$ | | |
| *Reasoning Models* | | | | | | | | |
| *o4-mini (high)*[*] | 94.9% | 78.2% | 21.6% | $-2.5\%$ | $-6.6\%$ | $-10.2\%$ | **70.3%** | 0.0269 |
| *gpt-5 (medium)*[*] | 89.5% | 77.6% | 18.8% | $-2.3\%$ | $-5.6\%$ | $-8.9\%$ | **67.7%** | 0.0390 |
| *o4-mini (medium)*[*] | 89.2% | 73.6% | 20.3% | $-2.7\%$ | $-8.8\%$ | $-11.8\%$ | **66.1%** | 0.0205 |
| *google/gemini-2.5-pro*[*] | 94.0% | 53.1% | 8.5% | $-2.8\%$ | $-7.2\%$ | $-9.3\%$ | **61.6%** | 0.2015 |
| *deepseek-v3.1 (thinking)* | 89.2% | 59.8% | 11.5% | $-2.0\%$ | $-4.1\%$ | $-8.1\%$ | **60.5%** | 0.0276 |
| *deepseek-r1* | 80.3% | 36.4% | 5.1% | $-3.2\%$ | $-6.8\%$ | $-7.3\%$ | **55.6%** | 0.0250 |
| *o3-mini (medium)*[*] | 86.2% | 50.0% | 6.0% | $-2.8\%$ | $-7.8\%$ | $-7.0\%$ | **55.1%** | 0.0230 |
| *qwen3-235b-a22b* | 80.2% | 39.7% | 5.1% | $-2.3\%$ | $-9.9\%$ | $-13.3\%$ | **53.5%** | 0.0343 |
| *gemini-2.5-flash*[*] | 81.4% | 22.6% | 4.8% | $-2.6\%$ | $-7.7\%$ | $-6.3\%$ | **47.7%** | 0.0090 |
| *grok-3-mini*[*] | 77.8% | 21.7% | 3.3% | $-2.1\%$ | $-10.0\%$ | $-6.4\%$ | **46.4%** | 0.0035 |
| *claude-3.7-sonnet*[*] | 76.2% | 24.1% | 2.4% | $-1.2\%$ | $-11.5\%$ | $-5.2\%$ | **45.5%** | 0.1282 |
| *Non-Reasoning Models* | | | | | | | | |
| *deepseek-chat-v3.1* | 82.7% | 29.3% | 3.9% | $-2.0\%$ | $-5.1\%$ | $-7.3\%$ | **49.8%** | 0.0068 |
| *gpt-4.1-mini*[*] | 73.7% | 20.9% | 3.8% | $-2.8\%$ | $-8.8\%$ | $-7.4\%$ | **42.4%** | 0.0070 |
| *gpt-4.1*[*] | 62.1% | 21.8% | 1.4% | $-3.0\%$ | $-10.2\%$ | $-10.4\%$ | **36.5%** | 0.0071 |
| *qwen3-coder* | 66.5% | 9.3% | 0.0% | $-1.3\%$ | $-10.5\%$ | $-9.9\%$ | **35.4%** | 0.0145 |
| *claude-sonnet-4*[*] | 60.7% | 14.0% | 2.0% | $-1.5\%$ | $-13.1\%$ | $-5.1\%$ | **32.4%** | 0.0211 |
| *llama-4-maverick* | 51.3% | 8.6% | 0.0% | $-1.3\%$ | $-9.5\%$ | $-8.8\%$ | **26.2%** | 0.0006 |
| *gpt-4o*[*] | 31.3% | 2.2% | 0.0% | $-0.4\%$ | $-2.2\%$ | $-6.4\%$ | **15.4%** | 0.0139 |
| *qwen-2.5-32b-coder* | 27.2% | 2.2% | 0.0% | $-1.8\%$ | $-0.3\%$ | $-5.5\%$ | **13.4%** | 0.0038 |
| *gemma-3-27b-it* | 26.1% | 2.2% | 0.0% | $-0.2\%$ | $-4.3\%$ | $-3.5\%$ | **13.1%** | - |
| *llama-3.3-8b-instruct* | 11.2% | 1.1% | 0.0% | $-0.4\%$ | $-0.2\%$ | $-0.3\%$ | **5.5%** | 0.0002 |

*Table 1.* UniCode Leaderboard. We report Pass@1 across three difficulty levels and evaluate model robustness via $\Delta$ ($\Delta$ = Overall − w/o Test), where larger drops indicate greater vulnerability. The significant drops ($\Delta \downarrow$) demonstrate the UniCode's capability to expose reasoning flaws. The cost per problem is included to help identify cost-effective models for future research. (*: closed models).

**Stage 3: LLM Adjudication** If no majority exists, the top two outputs ($o_1, o_2$) are sent to a high-reasoning LLM (e.g., o4-mini) for analysis. If the LLM yields a decisive judgment, that output is accepted; otherwise, the input is discarded to ensure data integrity. We validate that each component improves test case accuracy in Table 3.

## 4. Benchmark Curation and Leaderboard

This section describes the construction and validation of the UniCode benchmark. We first present our data curation pipeline and a human study evaluating problem quality, followed by a comprehensive leaderboard overview.

### 4.1. Data Pipeline and Quality

**Problem Generation** We curated 25,000 seed problems from platforms like LeetCode and CodeForces, filtering for competitive quality and clear specifications. An LLM assigned hierarchical tags (e.g., *graph → shortest-paths*) to identify 1–3 core skills per task (App. A.6). Following §2, we leveraged *o4-mini* as the main generator and *deepseek-r1* as adjudicator, and generated augmented variations from 600 seeds across 15 algorithms. After excluding trivial problems solved by all baseline models, we successfully distilled a final set of 492 candidate problems. App. B.3 confirms

that model rankings remain consistent across different generators, mitigating potential self-preference bias.

**Test Suites and Constraints** Each problem includes five components: description $D$, tag set $\mathcal{T}$, time limit $TL$, memory limit $ML$, and test cases $\mathcal{C}$. The time limit $TL$ and memory limit $ML$ are determined by running validated, optimized solutions $\mathcal{O}_{\text{valid}}$:

$$\text{TL} = \left\lceil k \cdot \min_{o \in \mathcal{O}_{\text{valid}}} T(o) \right\rceil, \qquad \text{ML} = \left\lceil k \cdot \text{Mem}(o^\star) \right\rceil,$$

where $o^\star = \arg\min_{o \in \mathcal{O}_{\text{valid}}} T(o)$. We select the minimum runtime across validated reference solutions to avoid loose time limits and multiply it by a conservative safety factor $k = 3$ to accommodate variations in alternative correct implementations. We then execute $T(\cdot)$ and $\text{Mem}(\cdot)$ within a secure sandbox environment (Bytedance-seed et al., 2025).

**Test Suite Quality** We ensure benchmark rigor via a multi-tiered validation process. `Human Validation`: Expert review of 113 generated problems yielded a 98.2% validity rate with 92.3% inter-annotator agreement (App. B.4). We further audited "Extremely Hard" tasks where all LLMs failed, removing 9 invalid cases from the 115 examined, as they exhibited ambiguous demonstrations or flawed test

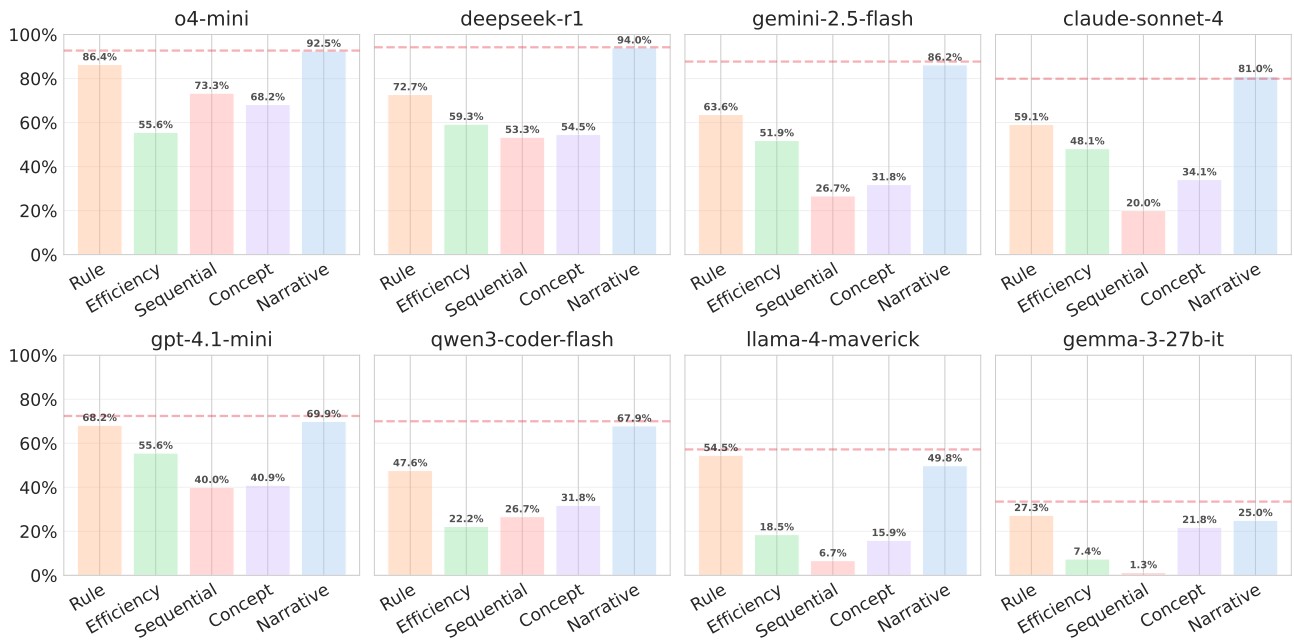

*Figure 3.* Model performance across reasoning variants. Red dashed lines denote seed problem performance. Key observations: (1) All models exhibit performance variance (e.g., claude-sonnet-4 shows a gap of $> 60\%$), suggesting substantial disparities in reasoning capabilities across different dimensions. (2) Even reasoning-optimized models (o4-mini, deepseek-r1) are vulnerable to sequential integration, concept fusion and efficiency scaling, which exposes a critical fragility of multi-step reasoning under novel constraints.

cases. `Automated Verification`: We verify the reliability of our stress-driven pipeline on *Test-Eval* (Yang et al., 2025b), an existing human-curated dataset. Our generated test cases achieved 94.5% correctness and 86.0% coverage (Table 3), significantly surpassing the baseline. `Theoretical Foundation`: While automated generation is not entirely error-free, our mathematical proof in App. B.5 establishes that the framework remains statistically robust for objective model evaluation.

**Release Artifacts.** We will release problem statements, test suites, and metadata (tags, generators, and prompts) to support reproducibility and downstream analysis.

### 4.2. UniCode Leaderboard

To offer a macroscopic perspective on the UniCode landscape, we evaluate 19 state-of-the-art LLMs across various architectures, parameter scales, and reasoning capabilities.

As shown in Table 1, UniCode is both highly challenging and discriminative, with overall pass@1 scores ranging from 70.3% (*o4-mini-high*) to 5.5% (*llama-3.3-8b-instruct*). Performance collapses on the hard split, where several models record 0.0%, underscoring the benchmark's difficulty. Reasoning-oriented models lead the rankings, validating the effectiveness of test-time compute scaling for complex logical inference. Our results show over 90% alignment with uncontaminated benchmarks (App. A.4), confirming that

UniCode is robust against data contamination and provides an unbiased assessment of model performance.

Models generally struggle more with adversarial and corner cases than random generation. For instance, *qwen3-235b-a22b* show pronounced sensitivity, with $\Delta_{\text{Corn}}$ reaching $-13.3\%$. Conversely, *deepseek-v3.1 (thinking)* exhibits superior stability with minimal performance drops. Furthermore, cost–performance analysis identifies *o4-mini (high)* as highly efficient, achieving top-tier pass@1 at $0.0269 per problem - nearly $7.5\times$ more cost-effective than *gemini-2.5-pro* for similar accuracy. While these results reveal UniCode's difficulty, they do not pinpoint specific failure modes. The following section decomposes performance across five reasoning axes to uncover where and why models fail.

## 5. In-Depth Code Reasoning Analysis

In this section, we investigate the fragility of LLM code reasoning across diverse variants and introduce a fine-grained taxonomy to categorize the origins of these failures.

### 5.1. How Fragile is Code Reasoning in LLMs?

To better understand the reasoning ability of LLMs, we curated 132 new problems using *livecodebench v1* [1] seed tasks, where models have previously excelled. We evaluated

---

[1] Initial Release: May 2023 – March 2024.

a selection of reasoning-specialized, general-purpose, and open-source models across various performance tiers.

**General Performance Drop and High Variance**  As shown in Figure 3, all models suffered significant performance declines, revealing their fragility when handling structural and conceptual shifts in reasoning tasks. We observe that even state-of-the-art LLMs fail to achieve comprehensive mastery across diverse reasoning paradigms. Furthermore, models exhibit non-negligible variance across test sets; for instance, *claude-3.5-sonnet* shows a performance gap exceeding 60% between scenarios. This imbalance suggests that aggregate scores often mask significant deficiencies in generalization.

**Failure Under Structural Reasoning Alterations**  While some LLMs remain robust against narrative perturbations, all models suffer sharp performance declines when the underlying reasoning graph topology is altered. Constraint modifications consistently degrade performance, highlighting the difficulty of preserving logical coherence under novel requirements. Efficiency scaling is particularly challenging: even the specialized *o4-mini* experiences a 37% drop, marking its weakest dimension. The most significant failures occur in sequential reasoning and concept fusion, indicating persistent difficulty in composing multiple logical components and maintaining long causal chains. Notably, *gemma-3-27b-it* scored a negligible 1.3% in sequential tasks, representing a near-total loss of functional capacity. Model performance exhibits a continuous declining trend as the augmentation depth increases (see App. A.5).

**Seed-problem Regression Phenomenon.**  We observe that models often default to original seed-problem logic rather than reasoning from updated task specifications (see Figure 5). This behavior suggests a reliance on heuristic shortcuts rather than rigorous logical deduction. For example, a model may employ a *simple parity count* because it recognizes a *palindrome sub-task*, yet fail to integrate new constraints. Alternatively, models rely on "low-efficiency templates" suitable for base problems without accounting for increased input scales. Future research should focus on enhancing models' zero-shot adaptation to novel constraints.

### 5.2. Where Do LLMs Fail in Code Reasoning?

To provide a more comprehensive overview of why LLMs fail, we introduce a systematic taxonomy to analyze their root causes. `Modeling error`: selecting an incorrect algorithmic paradigm for the task (e.g., opting for dynamic programming when a greedy approach suffices). `Logic bug`: implementation structural flaws (e.g., incorrect conditional or improper variable handling). `Indexing/caching bug`: incorrect cache siz-

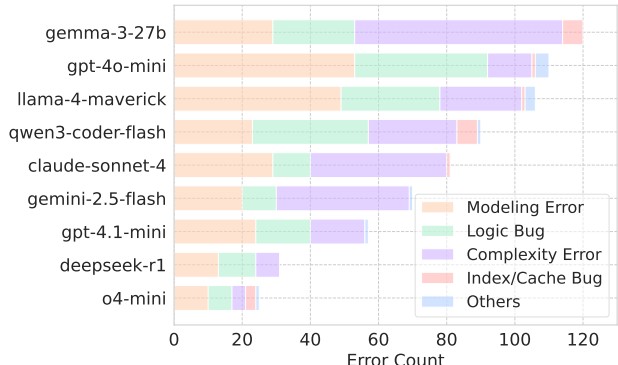

*Figure 4.* Error distributions across LLMs. Modeling and complexity errors are the primary failure modes, revealing inherent algorithmic weaknesses. (See Figure 5 for the detailed case study.)

ing, array bounds violations, or failed boundary checks. `Complexity error`: utilizing suboptimal algorithms when constraints demand more efficient solutions. `Others`: minor implementation oversights, including output formatting errors or library usage. We prompt LLMs to categorize errors by providing failed code, passed solutions and failed test cases (see App. Section B.9).

Through a fine-grained analysis of failed cases, we observe an "overhead" phenomenon: models like *gemma-3-27b-it* and *gpt-4o-mini* rarely fail due to isolated mistakes. Instead, they exhibit cascading failures where multiple errors occur simultaneously. This indicates that when a task exceeds a certain complexity threshold, the model's logical coherence may break down, resulting in repetitive, nonsensical code snippets or hallucinatory logic.

As shown in Figure 4, indexing and other errors account for the smallest fraction of failures. Although dominant error types vary, modeling errors remain a primary challenge, indicating persistent difficulty in problem conceptualization and algorithm selection. Another major contributor is complexity error: models such as *gemini-2.5-flash* and *gemma-3-27b-it* struggle with time complexity analysis and appear insensitive to resource constraints. These results suggest that while LLMs are proficient at syntax, they lack a robust grasp of algorithmic efficiency and conceptual modeling.

## 6. Related Work

**Competitive Coding**  LLM code generation evaluation is a rapidly evolving field (Jaech et al., 2024; Li et al., 2023; Guo et al., 2024; Hui et al., 2024; Zhang et al., 2023; Guo et al., 2025; Li et al., 2022a; Shao et al., 2024; Allal et al., 2023; Zhao et al., 2025), as code is increasingly viewed as a potential source of reasoning ability (Fu et al., 2022; Li et al., 2022c). Traditional benchmarks (Chen et al., 2021; Austin et al., 2021; Hendrycks et al., 2021) are grounded in static patterns, which have become increasingly vulnerable

**Problems**

P1: Check if a string can form a palindrome after removing $k$ characters.

P2: Find a subarray with strictly alternating.

New problem: Given an array, remove exactly $k$ elements such that the remaining $n$-$k$ elements can be rearranged into a palindrome with strictly alternating adjacent parities.

Concept fusion. New problem enforces simultaneous global symmetry (palindrome by value) and local structural constraints (parity alternation).

**Problems**

P1: Given an array nums of size $N$, find a **non-empty subset** with the maximum product.

$$N \leq 13, nums[i] \in [-9, 9].$$

New problem: Select a subset of exactly $K$ elements from nums to maximize their product.

$$1 \leq K \leq N \leq 200.$$

Efficiency scaling. The scale-up from $N = 13$ to $N = 200$ shifts the requirement from brute-force or greedy strategies to $DP$ with $O(NK)$.

**Failure Analysis**

**Modeling Errors**
GPT-4o and Gemini exhibit structural modeling failures and reduce the task to a simple parity-counting problem.

Error Pattern 1: Over-simplification

```
# ERROR: Treats it as only balancing even/odd counts
even_total = sum(1 for x in arr if x % 2 == 0)
odd_total = n - even_total
```

Error Pattern 2: Logic Inconsistency

```
# ERROR: Structural symmetry of palindromes is ignored
if even_required > even_total or odd_required > odd_total:
    return False
...
if even_to_remove + odd_to_remove == k:
    return True
```

**Failure Analysis**

**Complexity Errors**
Qwen-Flash and LLaMA produce logically correct but computationally infeasible solutions. They rely on itertools.combinations or naive DFS.

Error Pattern 1: Over-computation

```
# ERROR: Combinatorial explosion O(C(N, K))
for subset in combinations(nums, k):
    res = max(res, math.prod(subset))
```

Error Pattern 2: Over-computation

```
# ERROR: Exponential DFS O(2^N) without memoization
def dfs(idx, count, p):
    if count == k: return p
    if idx == n: return -float('inf')
    return max(dfs(idx + 1, count + 1, p * nums[idx]),
               dfs(idx + 1, count, p))
```

*Figure 5.* Case study of error patterns in code reasoning. Model failures are not stochastic but exhibit a *seed-problem regression*. Even when new problem invalidate original P1/P2 logic, LLMs frequently revert to memorized modeling paradigms or suboptimal complexities that were sufficient for the base problems. This suggests that LLMs often rely on heuristic shortcuts rather than true logical synthesis.

to data contamination and over-fitting through statistical shortcuts (Oren et al., 2023; Golchin & Surdeanu, 2023; Riddell et al., 2024; Roberts et al., 2023; Tang et al., 2024). While recent initiatives (Li et al.; Gu et al.; Zhu et al., 2025; Chambon et al.) integrate complex competitive programming tasks to stress-test algorithmic reasoning, they remain inherently static, resulting in delayed updates and a fixed set of problems that models can eventually memorize (Zheng et al., 2025b; Jain et al., 2024). This bottleneck underscores a critical need for a generative evaluation paradigm that dynamically scales problem complexity.

**Generative and Augmented Evaluation** Accurate algorithmic assessment requires rigorous problems and comprehensive test cases, traditionally necessitating manual curation (Chen et al., 2021; Hendrycks et al., 2021; Austin et al., 2021; Li et al.; Quan et al., 2025). While some studies leverage LLMs for test synthesis (Chen et al., 2022; Schäfer et al., 2023; Liu et al., 2023a; Wang et al., 2025; Jain et al., 2024), their problem designs typically adhere to fixed human-centric paradigms (Schäfer et al., 2023; Tufano et al., 2022; Chen et al., 2022; Liu et al., 2023b). Generative and augmented evaluation introduces dynamic tasks across diverse scenarios (Zheng et al., 2025a; Lin et al., 2025; Parmar et al., 2024; Zhu et al., 2023; Shi et al., 2025). Yet, current methods frequently lack the algorithmic depth and reasoning complexity required for advanced tasks (Chou

et al., 2025; Lops et al., 2025; Anand et al., 2013; Sofokleous & Andreou, 2008; Tufano et al., 2020; Swain et al., 2012). In this work, we generate evolving algorithmic variants through systematic operators. This paradigm provides rich diagnostic signals and, unlocks discovery potential akin to *Alpha-Evolve* (Novikov et al., 2025), uncovering seed-problem regression.

**Reasoning in LLMs** Whether LLMs possess genuine reasoning or perform sophisticated pattern matching remains debated (Wu et al., 2024; Hazra et al., 2025). Many argue their reasoning is fragile (Kambhampati, 2024; Gignac & Szodorai, 2024; Agrawal et al., 2025; Kim et al., 2024; von Recum et al., 2026), relying on data shortcuts (Wang et al., 2024) and sensitive to token bias (Jiang et al., 2024). While instruction-tuning (Xu et al., 2024; Luo et al., 2023) and recent benchmarks (Stolfo et al., 2023; Mirzadeh et al., 2024; Li et al., 2024a; Wang & Zhao, 2024; Yang et al., 2025a; Ramezanali et al., 2025; Orvalho & Kwiatkowska, 2025; Patel et al., 2024) probe these limits, they focus on surface perturbations or the linear extension of reasoning steps. In contrast, UniCode employs multi-dimensional augmentation operators to fundamentally disrupt the underlying reasoning graph (Pei et al., 2025; Huang et al., 2025; Wu et al., 2021), and create a more rigorous testbed to determine whether LLMs can perform reasoning rather than pattern memorization.

## 7. Conclusion

In this paper, we introduce UniCode, a novel generative framework designed to probe the reasoning boundaries of Large Language Models (LLMs) in code intelligence. To disrupt the reliance on statistical shortcuts, we implement multi-dimensional augmentations targeting structural, compositional, and conceptual shifts. This approach is supported by a scalable, stress-driven synthesis pipeline that ensures contamination-resistant evaluation. Our experiments reveal a 31.2% performance collapse across state-of-the-art LLMs, characterized by a "seed-problem regression", where models revert to memorized logic despite altered reasoning graphs. Additionally, high variance across reasoning axes challenges the reliability of traditional single-score benchmarks. Overall, this research underscores critical limitations in genuine code reasoning and highlights an urgent need for reasoning-oriented development in AI coding agents to bridge the gap between benchmarks and real-world applicability.

## Limitations and Future Work

While UniCode significantly reduces human burden in benchmark construction and reveals unique insights, several limitations remain. First, generating ground-truth test cases for complex, multi-step compositional problems remains a challenge, necessitating further research into more robust verification methods. Second, as code agents continue to evolve, there is a risk that models may "learn" the distribution of our augmentation axes, potentially leading to a new form of memorization (i.e., overfitting to the UniCode generation pipeline itself). Therefore, a critical future direction is to develop an evolvable, self-sustaining test generation pipeline that dynamically shifts its probing strategies, ensuring that the benchmark continues to challenge the evolving reasoning capabilities of next-generation models.

## Acknowledgement

This work was funded by the National Science and Technology Major Project (2022ZD0114902) and the National Natural Science Foundation of China (62376031). We thank Kewei Lian for his insights into the verification of coding problems. We are grateful to Dr. Chi Zhang for his guidance on the significance and limitations of this work, which provides a foundation for our future research. We thank Dr. Wenzheng Feng for his insightful writing guidance and support. Finally, I am deeply grateful to my daughter, Anan; her smiles have been my constant source of joy and strength during this challenging research journey. The road is long, but we continue to move forward.

## Impact Statement

This paper introduces UniCode, a framework designed to advance the field of machine learning by fostering genuine reasoning-oriented code intelligence. By exposing the "evaluation paradox" where models rely on statistical shortcuts, our work provides a critical foundation for developing more robust, generalizable AI agents. Beyond technical evaluation, UniCode contributes to the broader goal of building reliable and fair AI systems by systematically identifying logic fragilities. Ultimately, this research redefines benchmarking standards and facilitates the creation of safer AI technologies capable of handling complex, real-world reasoning tasks with higher fidelity and transparency.

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

# A. Extended Experimental Results and Diagnostics

## A.1. UniCode Efficiency Leaderboard

To illustrate the performance-cost efficiency frontier and assist users in selecting models that balance budget with performance, Figure 6 presents the UniCode leaderboard. This plot maps the Pass@1 score against the average cost per problem for various models, including: *gpt-5-2025-08-07, o4-mini-2025-04-16 high, o4-mini-2025-04-16 medium, gemini-2.5-pro, deepseek-v3.1-thinking, deepseek-r1-0528, o3-mini-2025-01-31, qwen3-235b-a22b, gemini-2.5-flash, grok-3-mini, claude-3.7-sonnet:thinking, deepseek-chat-v3.1, gpt-4.1-mini-2025-04-14, gpt-4.1-2025-04-14, qwen3-coder, claude-sonnet-4-20250514, llama-4-maverick:free, gpt-4o-2024-11-20, qwen-2.5-32b-coder*, and *llama-3.3-8b-instruct*.

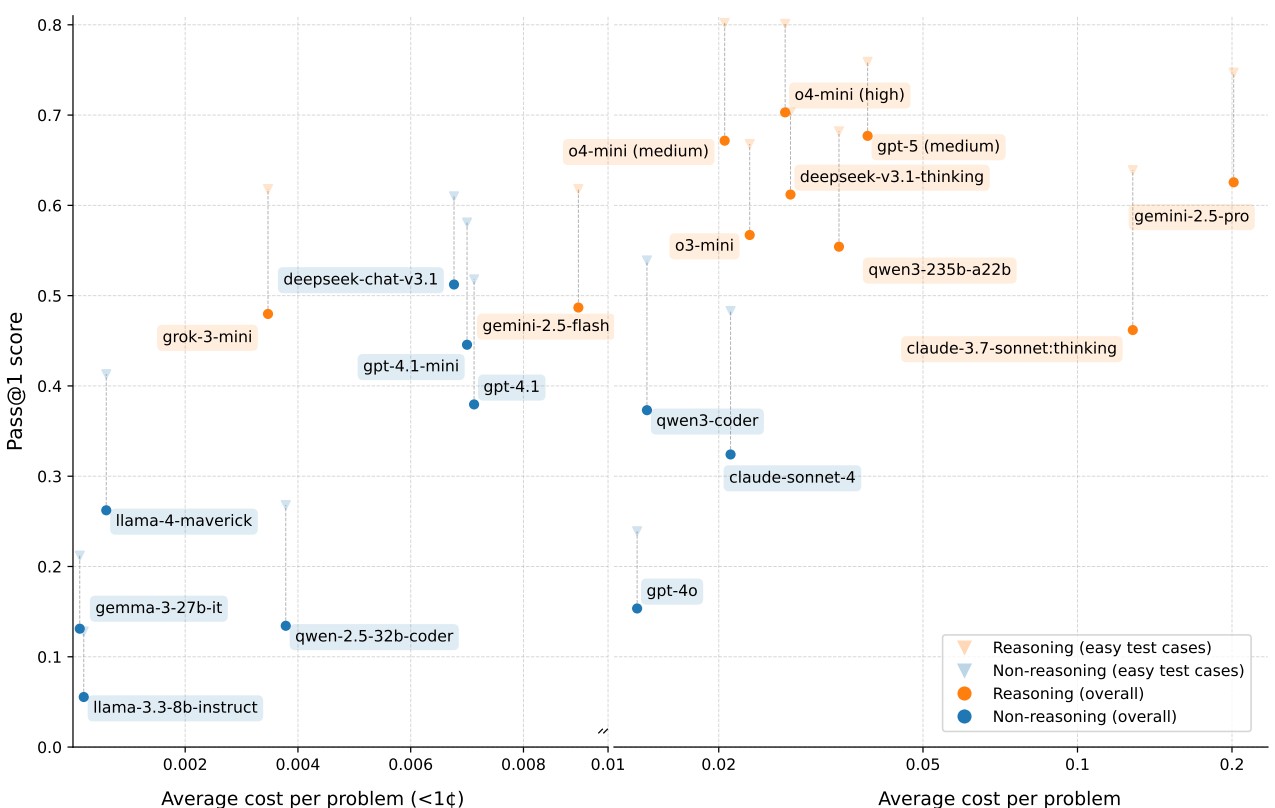

*Figure 6.* UniCode Leaderboard: Performance vs. cost efficiency across various models.

## A.2. Performance Across Algorithmic Paradigms

To better understand model capabilities, we sample representative models and categorize the problems by their primary algorithmic paradigm. Performance is evaluated using the pass@1 metric. As shown in Figure 7, the results reveal distinct strengths and weaknesses across various problem types.

The models demonstrate high proficiency in deterministic, template-driven tasks such as *fundamentals* and *data structures*. These problems, which often involve standard data structure manipulations, are likely well-represented in training corpora from textbooks and online repositories. Their solutions typically follow predictable patterns that models can easily recognize and reproduce. In contrast, performance drops on problems requiring novel reasoning and multi-step planning, such as *graph algorithms* and *dynamic programming* problems, which often necessitate customized logical deduction. This performance gap aligns with our earlier findings: a notable strength in template-driven tasks but a weakness in complex reasoning.

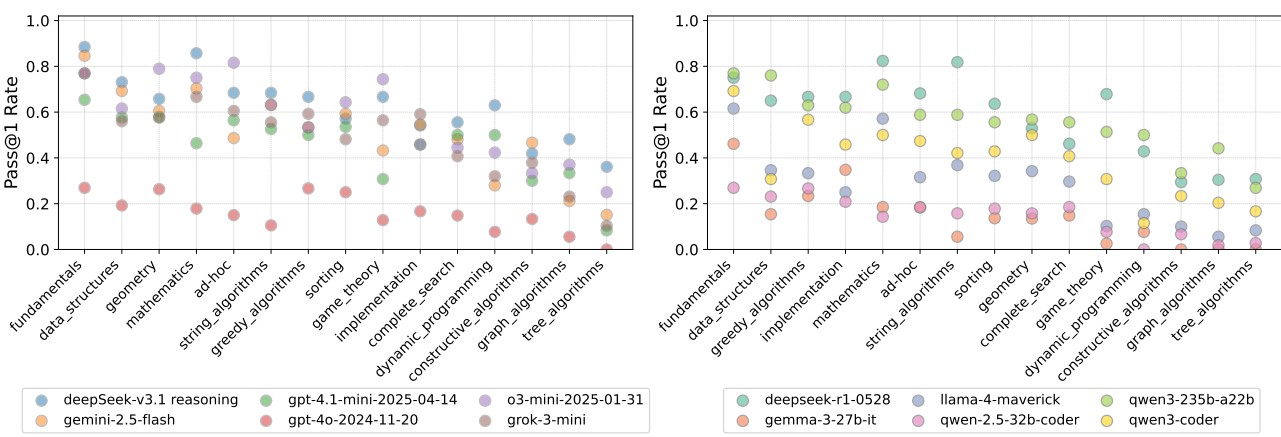

*Figure 7.* Comparative performance across problem types for closed-source (left) and open-source (right) models. The x-axis represents algorithmic tags. The y-axis refers to the Pass@1 rate. Best viewed zoomed in.

## A.3. Performance across Pass@k Settings

Since we generate the dataset using `o4-mini-medium`, we study the *pass@k* performance directly on it. *Pass@k* metric defines a problem as solved if any of the top-$k$ generated candidates passes all test cases. The results are shown in Table 2.

*Table 2.* `o4-mini (medium)` performance for *Pass@k* settings

| k | 1 | 2 | 3 | 4 | 5 | 6 | 7 | 8 | 9 | 10 |
|---|---|---|---|---|---|---|---|---|---|----|
| **Pass Rate (%)** | 66.1 | 73.6 | 77.4 | 79.8 | 81.4 | 82.5 | 82.8 | 83.5 | 83.5 | 83.5 |

The pass rate rises from 66.1% at $k = 1$ to 83.5% at $k = 10$, an improvement of over 20 percentage points. This indicates that repeated attempts significantly enhance performance, suggesting that many problems require multiple sampling to solve. The performance plateaus after $k = 8$, implying diminishing returns beyond this point. Even with multiple attempts, the model does not achieve a near-perfect score, illustrating the benchmark has a high upper limit and diagnostic value. These results underscore the importance of sampling numerous candidates for difficult tasks and reflect the complexity and variability inherent in the problems.

To analyze the intrinsic difficulty and recoverability of different problem categories, we group algorithm tags into three classes based on the marginal gain in pass rate from pass@1 to pass@3:

- **Significant Improvement ($\geq 20\%$).** This category includes tree algorithms, graph algorithms, and mathematical problems. These tasks often admit multiple valid solution paths or implementation strategies; consequently, sampling multiple candidates substantially increases the probability of success.

- **Moderate Improvement ($10\% \sim 20\%$).** This group comprises tags such as string algorithms and data structures. These problems typically follow established algorithmic templates, where additional sampling mainly helps mitigate localized implementation errors or *off-by-one* bugs.

- **Minor Improvement ($\leq 10\%$).** This category features dynamic programming and greedy algorithms. The limited benefit of increased sampling suggests that these problems either have a high baseline pass rate or rely primarily on rigorous structural reasoning rather than implementation variability.

The disparity in improvement across categories indicates that multi-candidate evaluation is highly effective for problems with diverse solution pathways, but yields diminishing returns for tasks that require deep logical reasoning or systemic correctness.

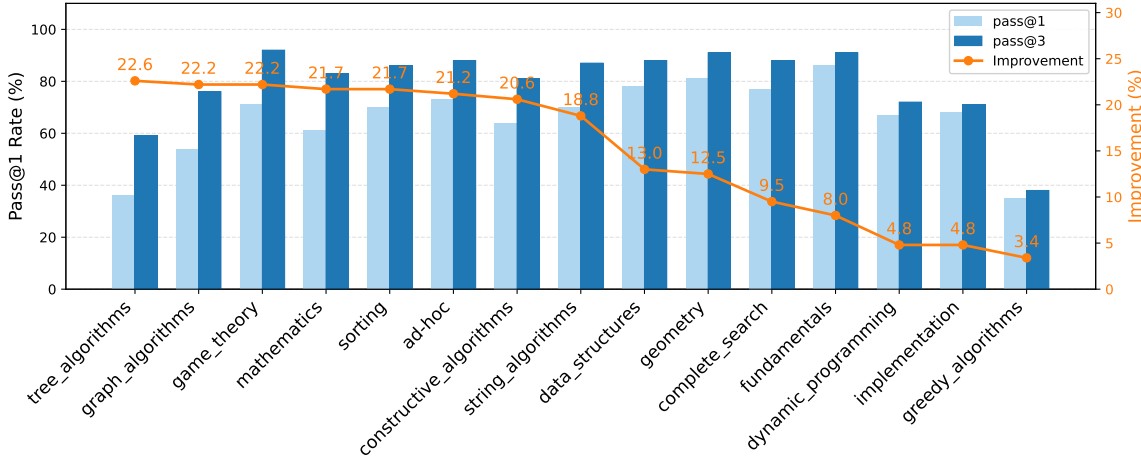

*Figure 8.* Per-tag improvement from *Pass@1* to *Pass@3*. Tags are grouped by improvement magnitude to illustrate which problem classes benefit most from candidate diversification.

### A.4. Alignment with Contamination-Free Code Benchmarks

To verify the validity and robustness of **UniCode**, we evaluate its alignment with two widely recognized, contamination-free benchmarks: **LiveCodeBench** and **LiveCodeBenchPro**. These benchmarks are specifically designed to mitigate the effects of data leakage, providing a reliable gold standard for performance comparison. To ensure experimental integrity, we utilized a consistent ensemble of representative models across all evaluations and conducted a **Pearson correlation analysis** to quantify the relationship between UniCode and these established metrics.

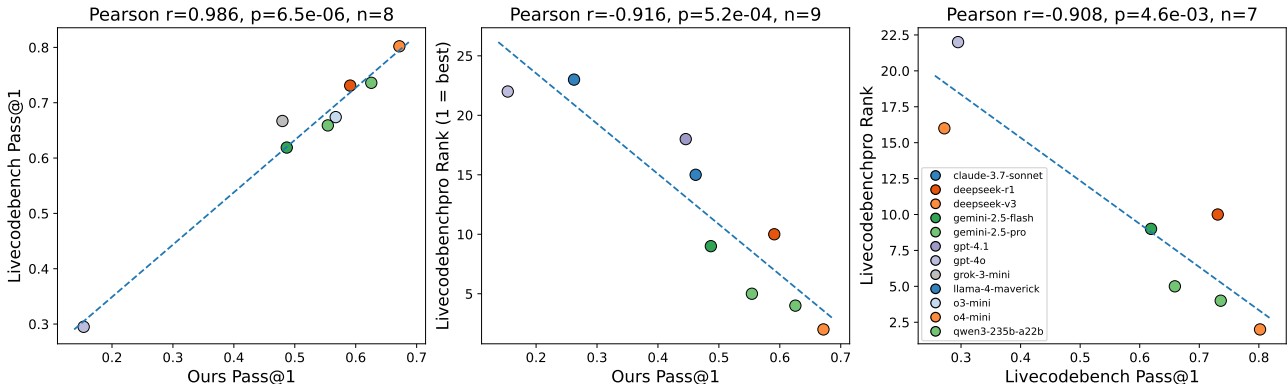

*Figure 9.* The alignment between UniCode and established benchmarks (LiveCodeBench and LiveCodeBenchPro). The degree of alignment we achieve, with reference to the absolute value of the correlation coefficient $r$, surpasses the inter-correlation among the established benchmarks.

The experimental results, as visualized in Figure 9, reveal a high degree of statistical consistency: we observe a strong positive correlation with LiveCodeBench ($r > 0.9$), indicating that UniCode's evaluation results are highly congruent with the scoring mechanisms of LiveCodeBench, and a strong negative correlation with LiveCodeBenchPro ($r < -0.9$), which is expected given the differing scoring conventions—while UniCode follows a "higher-is-better" metric, LiveCodeBenchPro employs a ranking-based system where lower numerical values indicate superior performance.

The high absolute correlation values ($|r| > 0.9$) across both benchmarks provide empirical evidence that **UniCode** serves as a reliable and high-fidelity proxy for model performance. This alignment confirms that our benchmark effectively captures the underlying coding capabilities of models.

## A.5. Performance Stability via Recursive Augmentation

To further investigate the depth and resilience of Large Language Models' (LLMs) reasoning capabilities, we conducted a recursive augmentation experiment. This process introduces cumulative structural and logical shifts, moving the test cases further away from the data distributions potentially encountered during pre-training.

**Experimental Setup** We randomly sampled a subset of 132 problems from Variant-L1 and re-applied the UniCode framework to generate 68 Variant-L2 problems. The evaluation spans three progressive levels:

- **Seed**: Original problems curated from human-centered benchmarks (e.g., Codeforces).

- **Variant-L1**: First-generation variants generated based on the Seed problems.

- **Variant-L2**: Second-generation variants generated by applying UniCode's augmentation logic to Variant-L1.

**Findings and Analysis** As illustrated in Figure 10, a consistent performance collapse is observed across all tested models. While models exhibit varying degrees of resilience at L1, the transition to L2 leads to an additional average decline of 7.9%.

Interestingly, the performance drop from L1 to L2 is generally less severe than the initial collapse from Level 0 to L1. We posit that the transition from Seed to L1 primarily serves to decouple statistical shortcuts and rote memorization of canonical training data. Once these shortcuts are neutralized at L1, the subsequent move to L2 tests the model's intrinsic reasoning depth rather than further memory exploitation.

Specifically, the results reveal a divergence in model robustness: Models like `o4-mini` demonstrate a resilient reasoning core, with a marginal decline of only 2.6% from L1 to L2. This suggests its internal logic remains stable despite increased complexity. In sharp contrast, `Gemini-1.5-Flash` experiences a substantial drop of 19.4%, indicating that its performance is highly sensitive to even minor structural perturbations once canonical patterns are removed.

**Conclusion** This recursive evaluation confirms that current static paradigms severely overestimate model intelligence by conflating memorization with reasoning. UniCode's ability to generate deeper-level variants provides a more realistic and rigorous "upper bound" for evaluating the genuine logical capacities of LLMs.

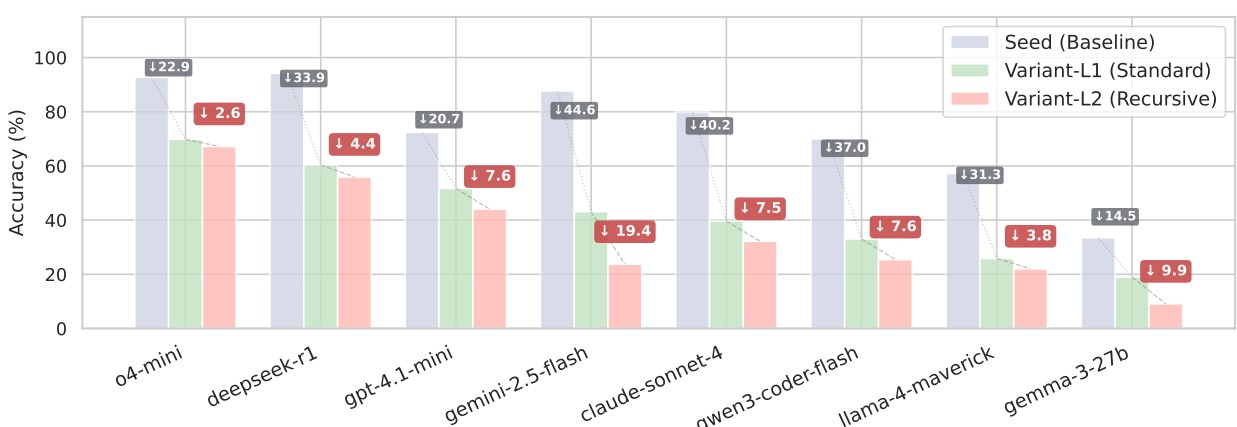

*Figure 10.* Robustness Decay under Recursive Augmentation. Accuracy of various LLMs on the original seed problems (Seed), first-order variants (Variant-L2), and second-order variants (Variant-L2) generated by UniCode.

## A.6. Code-tag Distribution

To systematically evaluate code generation capabilities of large language models (LLMs), we constructed a hierarchical taxonomy that organizes algorithmic knowledge into tags, subtags, and atomic skills. In total, the taxonomy consists of **9 top-level tags**, **31 subtags**, and **161 skills**, covering both fundamental algorithms and advanced techniques. Figure 11 provides a summary of this distribution.

This taxonomy brings several advantages for code generation evaluation. First, it ensures broad algorithmic coverage: the tags span essential paradigms such as graph algorithms, dynamic programming, data structures, and mathematical methods, allowing evaluations to probe diverse coding skills. Second, the inclusion of fine-grained subtags and skills provides diagnostic granularity. Instead of producing only aggregate scores, we can profile model performance across different algorithmic domains, exposing specific strengths (e.g., string hashing, greedy heuristics) and weaknesses (e.g., bitmask dynamic programming, numerical stability). Third, tagged organization supports balanced dataset construction, ensuring that evaluations are not biased toward a narrow set of skills. It also facilitates longitudinal comparisons: since the taxonomy is stable, we can track progress across model iterations and architectures. Finally, many of the listed skills, such as hashing and network flow, are directly relevant to industrial software engineering and competitive programming, thereby improving the real-world applicability of the evaluation.

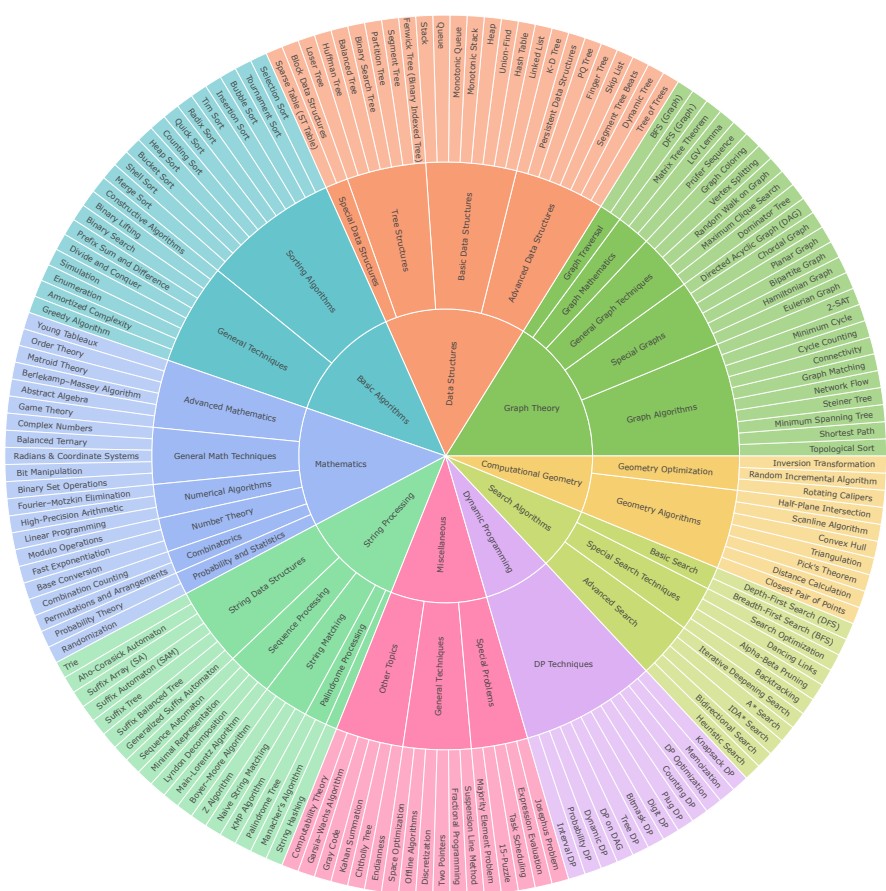

*Figure 11.* Distribution of tags, sub-tags, and skills in UniCode dataset. Best viewed when zoomed in.

## A.7. Example Problems

SEED PROBLEM 1: PATH OF TASTY DISHES

**Problem Statement:** Read problems statements in Mandarin Chinese and Russian. Suraj, the Chief Prankster is back in action now and this time he has stolen the valentine's day gift given by Ashi (the love of Chef) to the Chef and ran away with it to Byteland.

Byteland is not a regular place like Chef's town. The safest way from Chef's town to Byteland is through the path of tasty dishes. The path is named so because there are magical tasty dishes which appear to the traveler that no one can resist eating. Also, Suraj has added a strong sleep potion to each of the dish on this path to stop anyone from following him.

Knowing the devilish nature of Suraj, Ashi is concerned about the Chef and has asked all of Chef's town people to help. The distance from Chef's town to Byteland through the path of tasty dishes is $X$ units. They have the location where the magic dishes are and how many people are required to eat it completely. Anyone who eats a dish would go to a long sleep and won't be able to continue. They have the information about the tribal clans that live along the path of tasty dishes who can be of real help in this journey.

The journey of Chef and his friends can be described as follows: There is a total of $B$ dishes on the path. Each dish is located at distance $x_i$ ($x_{i-1} < x_i$). To minimize the number of friends Chef has to leave behind, all of them have decided that exactly $y_i$ of them will eat the $i^{th}$ dish. Also, there are $C$ tribal chef clans. Clan $i$ is located at distance $p_i$ ($p_{i-1} < p_i$) with a population of $r_i$. If a group of at least $q_i$ men approaches them, they will join the forces.

**Input Format:**

- The first line contains an integer $T$, the number of test cases.
- Each test case contains:
    - Line 1: $X$ (distance to Byteland).
    - Line 2: $B$ (number of dishes).
    - Line 3: $B$ pairs of space-separated integers $x_i, y_i$.
    - Line 4: An integer $C$, followed by $C$ space-separated triplets $p_i, q_i, r_i$.

**Output Format:**  For each test case, print the minimum size of the group (including Chef) needed to reach Byteland.

**Constraints:**

- $1 \le T \le 10, \quad 1 \le X \le 10^9, \quad 1 \le B \le 10000$

- Subproblem 1 (25 pts): $C = 0$

- Subproblem 2 (75 pts): $1 \le C \le 10000$

- $1 \le x_i < X, \quad 1 \le p_i < X$ (all positions are distinct)

- $1 \le y_i, q_i, r_i \le 10^{14}$

SEED PROBLEM 2: BARATHEON'S REIGN

**Problem Statement:** The Baratheons have been ruling in the Seven Kingdoms for many years. King Joffrey Baratheon commanded to build two monuments. The Baratheons have been ruling for $N$ years. Every year is described by an integer $A_i$, the level of prosperity.

You are to pick two historical periods $[S_1, F_1]$ and $[S_2, F_2]$ with the following rules:

- **No overlap:** Two periods shouldn't have common years.

- **Chronological:** The first period must start earlier than the second one.

- **Separation:** There must be at least $K$ years between $F_1$ and $S_2$.

Goal: Maximize the total sum of prosperity levels in the chosen periods.

**Input Format:**

- Line 1: $T$ (test cases).
- Each case:
    - Line 1: $N$ (years) and $K$ (gap).
    - Line 2: $N$ integers $A_1, A_2, \ldots, A_N$.

**Output Format:** For each test case, output a single line containing the maximum sum.

**Constraints:**

- $1 \leq T \leq 5, \quad 2 \leq N \leq 10^5, \quad 0 \leq K \leq 10^5$
- $-10^9 \leq A_i \leq 10^9, \quad K + 2 \leq N$

NEW PROBLEM: CHEF'S GRAND EXPEDITION

**Problem Statement:** Chef must journey in two phases.

**Phase 1: Recruitment**

- There are $N$ districts in Chef's town, labeled $1 \ldots N$. District $i$ has $A_i$ potential volunteers ($A_i$ may be negative: a negative value means the district actually shuns the effort).
- Chef may conduct exactly two recruitment campaigns, each on a contiguous interval of districts $[L, R]$. These two intervals must not overlap, and there must be at least $K$ districts between the end of the first and the start of the second.
- Chef gathers the sum of $A_i$ in each chosen interval. His total recruits $H$ is the sum over both intervals (if that sum is negative, he would of course choose intervals giving non-negative sum).

**Phase 2: Expedition**

- The path from Chef's town to Byteland has $B$ magical "dishes" at strictly increasing distances $x_i$. To cross dish $i$, exactly $y_i$ members of Chef's party must stop (and thus be lost to sleep).
- There are $C$ tribal clans at strictly increasing distances $p_j$. Clan $j$ will join Chef's party and contribute $r_j$ people, but only if at the moment Chef arrives at $p_j$ his current party size is at least $q_j$.

Chef starts Phase 2 with $G_0 + H$ people, where $G_0$ is the size he sets aside before recruitment. As he moves in increasing order of position he encounters dishes and clans. He must ensure that at every dish he has $\geq y_i$ people (to send them to sleep) and that after subtracting $y_i$, his party remains $> 0$. Similarly, at each clan he gains $r_j$ if current $\geq q_j$.

Goal: Compute the minimal $G_0$ such that Chef can complete Phase 2 alive.

**Input Format:**

- Line 1: $T$ (number of test cases).
- For each test case:
    - Line 1: $N$ and $K$.
    - Line 2: $N$ space-separated integers $A_1, A_2, \ldots, A_N$.
    - Line 3: $B$ (number of dishes).
    - Next $B$ lines: Two integers $x_i, y_i$ for the $i$-th dish.
    - Next line: $C$ (number of clans).
    - Next $C$ lines: Three integers $p_j, q_j, r_j$ for the $j$-th clan.

**Output Format:**  For each test case, print one integer: the minimum $G_0$.

**Constraints:**

- $1 \leq T \leq 5, \quad 1 \leq N, B, C \leq 10^5, \quad 0 \leq K < N$

- $-10^9 \leq A_i \leq 10^9$

- $1 \leq x_i < x_{i+1} \leq 10^9, \quad 1 \leq y_i \leq 10^{14}$

- $1 \leq p_j < p_{j+1} \leq 10^9, \quad 1 \leq q_j, r_j \leq 10^{14}$

# B. Technical Implementation and Reproducibility

## B.1. Test Cases Quality and Ablation Study

We evaluate test suites using two metrics: *correctness* (accepting valid solutions) and *coverage* (rejecting invalid ones). An ideal test suite optimizes both axes simultaneously. Let $S_{\text{correct}}$ and $S_{\text{incorrect}}$ be sets of correct and incorrect submissions. A test suite $M$ passes a submission $s$ if $s$ succeeds on all test cases $m_i \in M$, denoted $\text{pass}(s, M) = 1$. We define:

$$\text{Corr@N} = \frac{|\{\, s \in S_{\text{correct}} \mid \text{pass}(s, M) = 1 \,\}|}{|S_{\text{correct}}|}, \tag{1}$$

$$\text{Cov@N} = \frac{|\{\, s \in S_{\text{incorrect}} \mid \text{pass}(s, M) = 0 \,\}|}{|S_{\text{incorrect}}|}. \tag{2}$$

*Table 3.* Quality evaluation of generated test suites. We compare our full pipeline against the rStar-Coder baseline (Liu et al., 2025) and perform an ablation study to quantify the contribution of each component. **Stage 1 (Brute-Force)** uses brute-force solvers on small random inputs. **MajVote (Unfiltered Solvers)** applies a majority vote to all generated solutions without pre-filtering. **MajVote (Filtered Solvers)** uses only solutions that passed the Stage 1 stress test. Our **Full Pipeline** integrates all stages and input types. "Validated suites" is the percentage of problems for which a suite passed all validation checks.

| | Method | Input types | Correctness | Coverage | Problems with validated suites (%) |
|---|---|---|---|---|---|
| rStar-Coder | *MajVote (Unfiltered)* | *rand* | 86.9% | 80.2% | 94.3% |
| Ours | *Stage 1 (Brute-Force)* | *rand* | 91.9% | 81.5% | 98.2% |
| | *MajVote (Unfiltered)* | *all* | 86.7% | 85.2% | 93.9% |
| | *MajVote (Filtered)* | *all* | 93.8% | 84.3% | 92.8% |
| | ***Full Pipeline*** | *all* | **94.5%** | **86.0%** | **94.8%** |

**Setup.** We evaluated 80 problems from the *Test-Eval dataset* (Yang et al., 2025b), each having an average of 200 pass/fail solutions, using test suites of size $N = 50$ (see Section B.2 for test suite composition). Note that our approach to majority voting differs from rStar-Coder in its granularity. While rStar-Coder aggregates at the solution level and discards a problem unless a majority of solutions exhibit identical input–output behavior; our aggregation is performed per test case, so disagreement on individual cases does not invalidate the entire problem.

**Results and Ablation Analysis.** As shown in Table 3, our full pipeline significantly outperforms the baseline in both correctness (94.5% vs. 86.9%) and coverage (86.0% vs. 80.2%). This performance gain is dissected through a series of ablation experiments:

- **Impact of Stage 1 Filtering:** The brute-force (BF) filter is essential for correctness, providing a +7.1% improvement (from 86.7% to 93.8%). Unlike optimized solvers prone to "seed regression", BF oracles use exhaustive search to avoid shared logical flaws. This significantly mitigates correlated failures (where multiple models share the same logical flaw), reducing the failure rate from 18.7% to 5.2% and thereby enhancing the reliability of the subsequent model consensus stage.

- **Contribution of Adversarial Inputs:** The transition from *Stage 1* (random inputs) to the *Full Pipeline* highlights the role of adversarial and corner inputs in enhancing coverage (increasing from 81.5% to 86.0%), proving their efficacy in exposing complex-case failures.

- **Robustness of Aggregation:** Despite the increased complexity, our per-test-case adjudication maintains a high "validated suite" rate (94.8%), outperforming rStar-Coder's rigid solution-level majority vote.

While the automatic generation system cannot be error-free, our analysis in App. B.5 confirms that the resulting benchmark remains statistically reliable for evaluating code generation.

## B.2. Test-Suite Composition and Parameter Selection

In this section, we provide a detailed justification for the composition of the final test suite $S$, which consists of 50 test cases with a fixed distribution: 20 random ($I_{\text{rand}}$), 20 adversarial ($I_{\text{adv}}$), and 10 corner ($I_{\text{corn}}$) inputs. This configuration was determined through an extensive empirical evaluation aimed at balancing correctness and coverage.

**Experimental Setup**   We conducted a hyperparameter sweep over multiple test suite compositions, evaluating each configuration on a held-out set of 48 problems with 960 human-crafted solutions. Each configuration was assessed using two key metrics:

- **Correctness**: the proportion of valid solutions that pass all test cases.

- **Coverage**: the proportion of invalid solutions that are correctly rejected.

*Table 4.* Representative sweep results across different distributions.

| Distribution | Correctness (%) | Coverage (%) |
|---|---|---|
| (5, 5, 0) | 97.9 | 77.0 |
| (10, 10, 5) | 95.8 | 81.4 |
| **(20, 20, 10)** | **94.0** | **87.5** |
| (30, 30, 20) | 91.7 | 88.7 |
| (50, 50, 20) | 91.7 | 90.0 |

We observe that smaller test suites (5,5,0) achieve high correctness but suffer from low coverage, failing to detect many faulty solutions. Larger suite such as (30,30,20), improves coverage marginally but at the cost of increased sensitivity to corner test cases and higher computational cost. The configuration (20, 20, 10) is selected as the optimal "elbow point. It maintains high correctness of 94.0% while achieving broad coverage of 87.5%, providing a rigorous filtering mechanism without being prohibitively expensive or overly punitive to valid code.

## B.3. Analysis of Generator Bias

A potential concern in generative evaluation is the risk of "generator bias", where a model performs better on problems it generated itself. To address this concern, we performed validation using an alternative, open-source generator `deepseek-r1` to generate a new set of 104 problems across 5 distinct tags. We then benchmark 6 models of varying capability levels on this independently generated set and compared the results to their performance on the standard `o4-mini`-generated UniCode problems and human-curated no data contamination LiveCodeBench[2].

| Model | Unicode (multi-mixed) | Unicode (deepseek-gen) | Unicode (o4mini-gen) | LiveCodeBench (human-curated) |
|---|---|---|---|---|
| *gpt-5* | 68.8% | 72.5% | 67.7% | — |
| *o4-mini* | 67.7% | 70.2% | 66.9% | 74.2% |
| *deepseek-r1* | 60.0% | 61.6% | 56.6% | 73.1% |
| *o3-mini* | 55.6% | 51.0% | 55.1% | 63.0% |
| *gpt-4.1-mini* | 44.5% | 41.3% | 42.4% | 53.2% |
| *gemma-3-27b-it* | — | 14.6% | 13.0% | — |

*Table 5.* Pass@1 rates across different problem generators. While absolute scores fluctuate, the relative model hierarchy remains highly consistent (Pearson $r = 0.984$).

As indicated in Table 5, the results do not support the presence of significant self-preference bias. The relative ranking of models remains highly consistent across all three datasets, with a Pearson correlation of $r = 0.984$ ($p = 4 \times 10^{-4}$) between model performances on the `deepseek-r1`-generated and `o4-mini`-generated problem sets.

---

[2]Initial Release: 8/1/2024 to 5/1/2025

Although `deepseek-r1` excels on its own problems (61.6% vs. 56.6% on `o4-mini`'s), a similar gap on Live-CodeBench suggests this reflects stylistic preferences rather than intentional bias. To mitigate such effects, we introduce `UniCode-Multi`, a composite benchmark aggregating problems from five diverse generators (`o4-mini`, `gpt-5`, `gemini-2.5-pro`, `Deepseek-r1`, and `qwen3-235b-a22b`). Results verify that this multi-source approach effectively smooths stylistic bias while maintaining consistent model rankings.

### B.4. Human Study

To rigorously assess the utility and complexity of our generated benchmark, we conducted an extensive human evaluation involving 113 problems (approx. 23% of the total problem set). This process required over 30 hours of expert labor, as each problem underwent 10–20 minutes of in-depth analysis by competitive programming veterans.

**Expert Annotation Protocol** We recruited five independent annotators: senior competitive programmers and algorithm engineers with over 5 years of experience and a Codeforces rating of 2100+ (Master level or above). To ensure objectivity, we employed a blinded rating protocol to evaluate the problems:

- **Validity & Rigor.** The evaluation yielded a high Validity rate of 98.2%. Technical analysis of the few invalid cases (1.8%) revealed that they were primarily due to minor output specification ambiguities rather than fundamental logical flaws or insurmountable constraints.

- **Inter-Annotator Agreement.** We observed a 92.3% agreement rate among the five experts. This level of consensus, especially given the problems' complexity, ensures the clarity and formal precision of the generated statements.

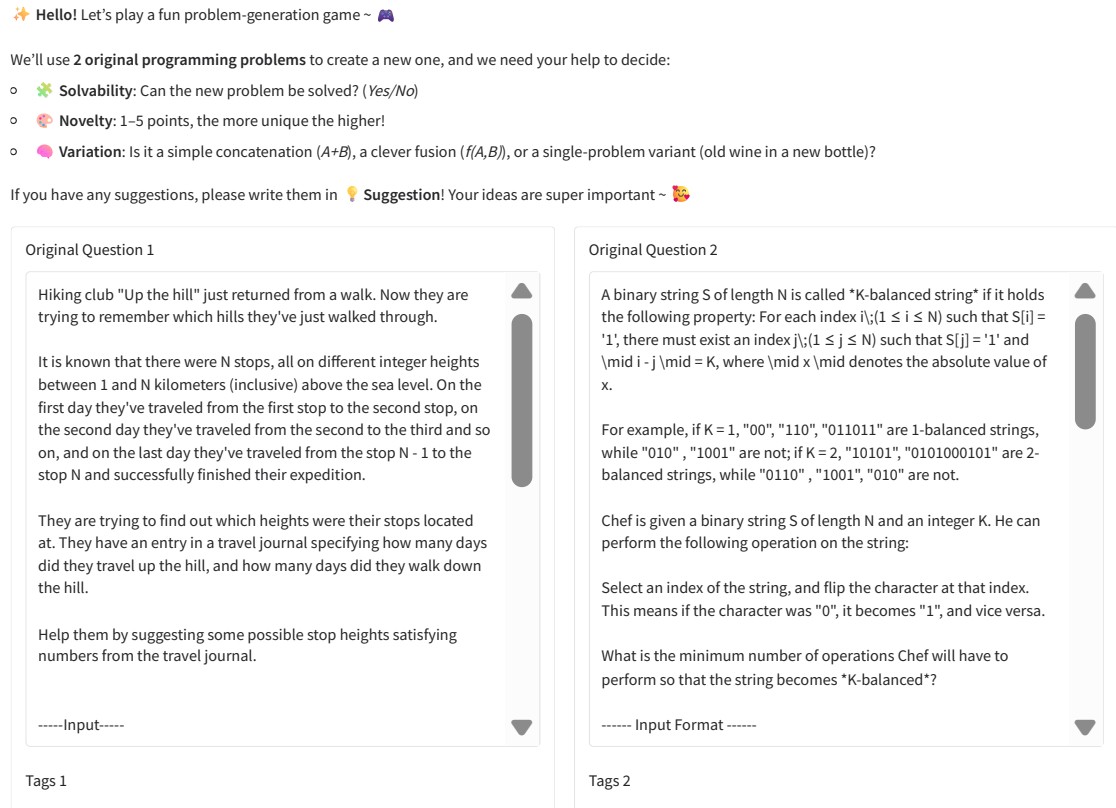

*Figure 12.* Human rating website. Best viewed when zoomed in.

['Implementation', 'Constructive algorithms']

['String algorithms', 'Constructive algorithms', 'Data structures', 'Ad-hoc']

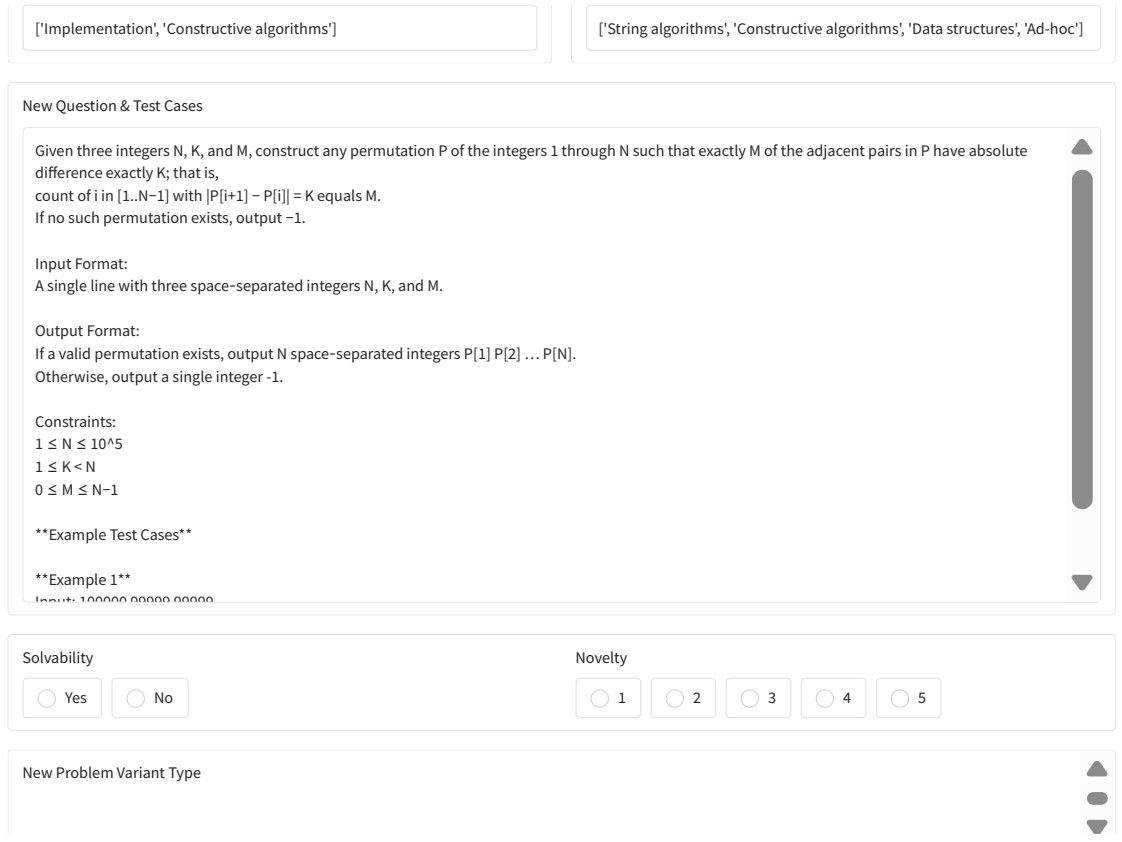

New Question & Test Cases

Given three integers N, K, and M, construct any permutation P of the integers 1 through N such that exactly M of the adjacent pairs in P have absolute difference exactly K; that is,
count of i in [1..N−1] with |P[i+1] − P[i]| = K equals M.
If no such permutation exists, output −1.

Input Format:
A single line with three space-separated integers N, K, and M.

Output Format:
If a valid permutation exists, output N space-separated integers P[1] P[2] … P[N].
Otherwise, output a single integer -1.

Constraints:
1 ≤ N ≤ 10^5
1 ≤ K < N
0 ≤ M ≤ N−1

**Example Test Cases**

**Example 1**
Input: 100000 99999 99999

Solvability

Yes   No

Novelty

1   2   3   4   5

New Problem Variant Type

*Figure 13.* Human rating website. Best viewed when zoomed in.

## B.5. Trustworthy Evaluation with Erroneous Tasks

Benchmarks for code generation occasionally contain *erroneous* items (e.g., unsolvable prompts, mislabeled I/O, flawed tests). This section develops a simple contamination model that quantifies how such items affect reported accuracy, provides bias- and variance-aware confidence bounds, and gives practical recipes to maintain trust in benchmark results.

### B.5.1. SETUP AND NOTATION

Let each task $i \in \{1, \dots, n\}$ be either *reliable* ($R_i = 1$) or *unreliable* ($R_i = 0$). Write

$$\alpha \equiv \Pr(R_i = 0) \quad \text{and} \quad 1 - \alpha \equiv \Pr(R_i = 1),$$

so $\alpha$ is the *contamination rate* of the benchmark. Let $p \in [0, 1]$ denote the model's true accuracy on reliable tasks and let $q_e \in [0, 1]$ denote the effective success probability on unreliable tasks (e.g., a random or spurious pass rate). For *pass@k*, define $p$ and $q_e$ analogously as the success probability within $k$ attempts.

For each task, the observed outcome $Y_i \in \{0, 1\}$ indicates success. The reported accuracy is $\hat{\mu} \equiv \frac{1}{n} \sum_{i=1}^{n} Y_i$.

### B.5.2. SYSTEMATIC BIAS (IDENTIFICATION AND DE-BIASING)

By the law of total expectation,

$$\mathbb{E}[\hat{\mu}] = (1 - \alpha)\, p + \alpha\, q_e \quad \Longrightarrow \quad \text{Bias}(\hat{\mu}; p) = \left| \mathbb{E}[\hat{\mu}] - p \right| = \alpha\, |q_e - p| \le \alpha. \tag{3}$$

Thus, when $\alpha$ is small, the systematic bias is small in absolute value. If $\alpha$ and $q_e$ are known (or fixed by design), an *unbiased* estimator of $p$ is obtained by de-biasing:

$$\tilde{p} = \frac{\hat{\mu} - \alpha q_e}{1 - \alpha} \qquad \text{(exact if } \alpha, q_e \text{ are known).} \tag{4}$$

When only bounds are available, $q_e \in [q_{\min}, q_{\max}]$ and $\alpha \in [0, \alpha_{\max}]$, one obtains a *conservative* identification region for $p$:

$$p \in \left[ \frac{\hat{\mu} - \alpha_{\max} q_{\max}}{1 - \alpha_{\max}}, \; \frac{\hat{\mu} - \alpha_{\min} q_{\min}}{1 - \alpha_{\min}} \right] \cap [0, 1]. \tag{5}$$

In practice, $q_{\max}$ can be set by a null-model baseline (e.g., trivial solver or random program generator), and $\alpha_{\max}$ by audit sampling.

### B.5.3. RANDOM ERROR (SAMPLING VARIABILITY)

There are two natural regimes for variance, depending on whether the reliable/unreliable split is fixed in advance (e.g., exactly $100(1 - \alpha)\%$ reliable) or arises by i.i.d. sampling.

**Fixed split (common in controlled curation).** If exactly $(1 - \alpha)n$ reliable and $\alpha n$ unreliable tasks are present,[3] then

$$\text{Var}(\hat{\mu}) = \frac{(1 - \alpha)\, p(1 - p) + \alpha\, q_e(1 - q_e)}{n}, \tag{6}$$

$$\text{SE}(\hat{\mu}) = \sqrt{\frac{(1 - \alpha)\, p(1 - p) + \alpha\, q_e(1 - q_e)}{n}}. \tag{7}$$

**Random mixture (i.i.d. contamination).** Marginally $Y_i \sim \text{Bernoulli}(\mu)$ with $\mu = (1 - \alpha)p + \alpha q_e$, hence

$$\text{Var}(\hat{\mu}) = \frac{\mu(1 - \mu)}{n} \qquad \text{with} \quad \mu = (1 - \alpha)p + \alpha q_e. \tag{8}$$

By concavity of $x(1 - x)$, (6) $\le$ (8), so using (8) is conservative when the split is fixed.

---

[3]Assuming $n(1 - \alpha)$ and $n\alpha$ are integers; otherwise interpret as the nearest integers.

**Confidence intervals.** Let $z_{0.975} \approx 1.96$. A simple large-sample 95% CI for $\mu$ is

$$\hat{\mu} \pm z_{0.975}\sqrt{\frac{\hat{\mu}(1 - \hat{\mu})}{n}}, \tag{9}$$

or, more accurately at small $n$, use a Wilson or Agresti–Coull interval for $\mu$. When $\alpha, q_e$ are *known*, a CI for $p$ follows from de-biasing:

$$\left[\frac{\underline{\mu} - \alpha q_e}{1 - \alpha}, \frac{\overline{\mu} - \alpha q_e}{1 - \alpha}\right], \tag{10}$$

where $[\underline{\mu}, \overline{\mu}]$ is a 95% CI for $\mu$. If only bounds are known ($\alpha \in [0, \alpha_{\max}]$, $q_e \in [q_{\min}, q_{\max}]$), combine (5) with $[\underline{\mu}, \overline{\mu}]$ to obtain a conservative CI for $p$:

$$\left[\frac{\underline{\mu} - \alpha_{\max}q_{\max}}{1 - \alpha_{\max}}, \frac{\overline{\mu} - \alpha_{\min}q_{\min}}{1 - \alpha_{\min}}\right] \cap [0, 1]. \tag{11}$$

B.5.4. TOTAL ERROR BOUND

Combining systematic and random components yields a high-probability bound on the absolute estimation error for $p$:

$$|\hat{\mu} - p| \le \underbrace{\alpha |q_e - p|}_{\text{systematic bias } \le \alpha} + \underbrace{z_{0.975}\sqrt{\frac{\mu(1 - \mu)}{n}}}_{\text{random error}} \quad \text{(w.h.p.)}. \tag{12}$$

When $n$ is large, the $O(n^{-1/2})$ term vanishes and the total error is controlled by the bias ceiling $\alpha$. If $\alpha$ (and $q_e$) are known, report the de-biased estimate (4) with CI (10); this both removes the bias and shrinks the CI.

B.5.5. STRATIFIED (TAG-WISE) CONTAMINATION

If tasks are grouped into tags $t = 1, \ldots, T$ with weights $w_t$ (sum to 1), reliable rates $p_t$, contamination rates $\alpha_t$, and unreliable success $q_{e,t}$, then

$$\mu = \sum_{t=1}^{T} w_t\big((1 - \alpha_t)p_t + \alpha_t q_{e,t}\big), \tag{13}$$

$$\text{Var}(\hat{\mu}) = \frac{1}{n}\sum_{t=1}^{T} w_t\big((1 - \alpha_t)p_t(1 - p_t) + \alpha_t q_{e,t}(1 - q_{e,t})\big), \tag{14}$$

under a fixed per-tag split. Reporting tag-wise de-biased estimates $\tilde{p}_t = (\hat{\mu}_t - \alpha_t q_{e,t})/(1 - \alpha_t)$ with their CIs, and then aggregating by the $w_t$, makes contamination assumptions explicit and auditable.

B.5.6. NUMERICAL ILLUSTRATION

Take $\alpha = 0.06$, $p = 0.80$, $q_e = 0.50$. Then

$$\mu = (1 - \alpha)p + \alpha q_e = 0.94 \cdot 0.80 + 0.06 \cdot 0.50 = 0.782, \quad \text{Bias} = |\mu - p| = 1.8\%.$$

Under a fixed split, the standard error is

$$\text{SE}(\hat{\mu}) = \sqrt{\frac{0.94 \cdot 0.80 \cdot 0.20 + 0.06 \cdot 0.50 \cdot 0.50}{n}} = \sqrt{\frac{0.1654}{n}}.$$

The resulting 95% CI half-width is $1.96 \times \text{SE}(\hat{\mu})$:

As $n$ grows, random error shrinks as $O(n^{-1/2})$; the residual error is then dominated by the (small) bias ceiling $\alpha$.

B.5.7. PRACTICAL SAFEGUARDS

- **Audit and bound $\alpha$.** Spot-check a random subsample to obtain an empirical upper bound $\alpha_{\max}$ with binomial CIs; report $p$ using (11).

*Table 6.* Random error and conservative total error bound (bias + half-width) at various $n$ ($\alpha$=6%, $p$=0.80, $q_e$=0.50).

| $n$ | SE | 95% CI half-width | Total error bound |
|---|---|---|---|
| 500 | 1.82% | 3.57% | $1.8\% + 3.57\% \approx 5.4\%$ |
| 5,000 | 0.575% | 1.13% | $1.8\% + 1.13\% \approx 2.9\%$ |
| 10,000 | 0.407% | 0.80% | $1.8\% + 0.80\% \approx 2.6\%$ |

- **Calibrate $q_e$.** Measure $q_e$ (or $q_{\max}$) using null models (e.g., trivial programs, permuted I/O) to cap spurious pass rates.

- **De-bias when possible.** If $(\alpha, q_e)$ are fixed by design (e.g., known faulty items), publish the de-biased estimate (4) and its CI (10).

- **Stratify and reweight.** Estimate per-tag $(\alpha_t, q_{e,t})$ and aggregate, reducing sensitivity to heterogeneous contamination.

- **Robust reporting.** Alongside $\hat{\mu}$, report (i) de-biased $\tilde{p}$, (ii) contamination-aware CIs, and (iii) sensitivity bands under $(\alpha, q_e)$ ranges as in (11).

**Takeaway.** Even when a benchmark contains a small fraction of erroneous tasks, its reported accuracy remains trustworthy when (i) contamination is explicitly modeled, (ii) bias is de-biased or bounded, and (iii) sampling error is controlled by adequate $n$. In the common regime of small $\alpha$ and large $n$, the total measurement error is tightly bounded and the benchmark reliably reflects true coding performance.

### B.6. Complexity Bounds and Time Constraints

To illustrate the relationship between input scale, algorithmic complexity, and execution time, we analyze two approaches to a standard programming problem:

- **Task**: Sort an array of $n$ integers and count inversions

- **Input Range**: $1 \leq n \leq 2 \times 10^7$

- **Expected Solutions**:
  - Optimal: Merge sort ($O(n \log n)$) with inversion counting
  - Suboptimal: Bubble sort ($O(n^2)$) with brute-force counting

#### B.6.1. CAPACITY ANALYSIS

We assume a typical modern computer can perform approximately $10^8$ operations per second. We set time limits as 5s for optimized and 50s for brute-force algorithms.

For the optimized algorithm ($T(n) = n \log_2 n$): we solve $n \log_2 n \leq 5 \times 10^8$ to show it can handle input $n = 2 \times 10^7$ in 5 seconds, as the number of operations, $2 \times 10^7 \times 24.2 \approx 4.84 \times 10^8$, stays within the $5 \times 10^8$ operations limit for 5 seconds at $10^8$ operations per second.

For the brute-force algorithm ($T(n) = n^2$): $n^2 \leq 50 \times 10^8$ yields $n \approx 7 \times 10^4$ maximum. In contrast, processing input $n = 2 \times 10^7$ would require $4 \times 10^{14}$ operations (around 46 days), demonstrating quadratic time growth.

*Table 7.* Algorithm Capacity Comparison

| Metric | Optimized ($O(n \log n)$) | Brute-force ($O(n^2)$) |
|---|---|---|
| Time Limit | 5s | 50s |
| Max $n$ | $2 \times 10^7$ | $7 \times 10^4$ |

The large difference (approximately 300x) in manageable input sizes ($2 \times 10^7$ vs $7 \times 10^4$) explains the stress-driven pipeline: the optimized algorithm verifies efficiency at competition-scale inputs, while the brute-force method allows small-case validation ($n \leq 10^4$ in $\leq 2$s). This setting ensures that the brute-force algorithm has enough time to pass test cases with smaller input sizes, which are usually used to verify basic correctness. This is very useful for debugging and initial testing.

## B.7. Integrated Prompts for Test Case Generation

*Listing 1.* Random Input Generator (CYaRon)

```
I will provide you with a programming problem description, and your task is to generate
    standardized test input samples using the CYaRon library.

You need to complete the following steps:
1. Parse the constraints on the input from the problem description, such as the range of
    input data, specific input constraints, etc.
2. Write a function generate_test_input using the CYaRon library to randomly generate
    test inputs based on a specified problem size. The function should validate that the
    parameters fall within the specified constraints. If any parameter is out of range,
    the function should return None. If the parameters are valid, generate a random test
    input and return an input string (input_string).
3. Write a function validate_test_input to verify whether the generated test input
    satisfies the requirements specified in the problem description. This includes
    checking the input data type and constraints parsed in step 1, such as range and
    other conditions. The function should take input_string as input and return a boolean
     (True/False).

Output format (strictly follow)
Part 1: Parse Input Constraints
Specify the input constraints as described in the problem.

Part 2: Code for Test Input Generation
import cyaron as cy #cyaron version: 0.7.0

def generate_test_input():
    # set parameters constraints that meet requirements (e.g. 1 <= N <= 300)
    ...
    # Generate input using CYaRon
    input_data = [
        ...
    ]
    return "\n".join(map(str, input_data))

Part 3: Code to Validate Test Input
def validate_test_input(input_string):
    # Validation logic
    return <boolean>

Note:
- cy.Integer() is not supported; it should be cy.randint.
- use cy.String.random instead of cy.String
- The function generate_test_input() should not accept any parameters. You need to
    generate the input entirely within the function.
- Generate code following the above format, without starting with ```python or similar
    markers.
```

*Listing 2.* Adversarial Input Generator (CYaRon)

```
I will provide you with a programming problem description, and your task is to generate
    adversarial test input samples using the CYaRon library.
```

```
You need to complete the following steps:
1. Parse the constraints on the input from the problem description, such as the range of
    input data, specific input constraints, etc.
2. Write a function generate_test_input using the CYaRon library to generate a single
    adversarial test input designed to challenge boundary conditions or worst-case
    complexity. The function should internally randomize which adversarial strategy to
    use, without accepting any parameters. The generated input should still conform to
    the problem constraints.
3. Write a function validate_test_input to verify whether the generated test input
    satisfies the requirements specified in the problem description. This includes
    checking the input data type and constraints parsed in step 1. The function should
    take input_string as input and return a boolean (True/False).

Output format (strictly follow):
Part 1: Parse Input Constraints
Specify the input constraints as described in the problem.

Part 2: Code for Test Input Generation
import cyaron as cy # cyaron version: 0.7.0
import random

def generate_test_input():
    # set parameters constraints that meet requirements (e.g. 1 <= N <= 300)
    ...
    # Randomly choose one adversarial strategy
    strategy = random.choice(["equal_weights", "alternating_large_small", "large_ends"])

    if strategy == "equal_weights":
        # example: all weights are maximal
        N = cy.randint(100000, 100000) # fix to worst-case size
        max_weight = cy.randint(109, 109)
        k = random.choice([1, N//2, N-1, N])
        weights = [max_weight] * N
    elif strategy == "alternating_large_small":
        N = cy.randint(100000, 100000)
        max_weight = cy.randint(109, 109)
        k = random.choice([1, N//2, N-1, N])
        weights = [max_weight if i%2 else 1 for i in range(N)]
    else: # large_ends
        N = cy.randint(100000, 100000)
        max_weight = cy.randint(109, 109)
        k = random.choice([1, N//2, N-1, N])
        weights = [max_weight] + [1]*(N-2) + [max_weight]

    # Build input string
    input_lines = [f"{N} {k}"] + [str(w) for w in weights]
    return "\n".join(input_lines)

Part 3: Code to Validate Test Input
def validate_test_input(input_string):
    try:
        lines = input_string.strip().split('\n')
        N_k = lines[0].split()
        if len(N_k) != 2:
```

```
            return False
        N, k = map(int, N_k)
        if not (1 <= N <= 100000):
            return False
        if not (1 <= k <= N):
            return False
        weights = list(map(int, lines[1:]))
        if len(weights) != N:
            return False
        for w in weights:
            if not (1 <= w <= 109):
                return False
        return True
    except:
        return False

Note:
- generate_test_input() must return a single adversarial input string, not a list.
- Use cy.randint instead of cy.Integer().
- The function should generate adversarial yet valid data fully inside, without
    parameters.
```

*Listing 3.* Direct Test Input Generator

```
Task:
 Generate a challenging test input for the algorithm problem:
 {problem_description}
 Instructions:
 - Focus on edge cases or scenarios that maximize the failure probability in faulty
     solutions.
 - Due to the output length limit, you should generate a small-scale test input that is
     complete and valid.
 - Output the test input directly, not code to generate it.
 Output format:
 '''plaintext
 {test input}
 '''
 Think step by step.
```

## B.8. Integrated Prompts for Algorithmic Problem Generation

*Listing 4.* Prompts for Compositional Variants

```
You are an expert competitive programmer.
I'll provide you with two programming problems.
If the problems test similar concepts (same-type fusion):
1. Analyze their problem design approaches
2. Create a new challenging problem testing the same concept(s).

If they test different concepts (cross-type fusion):
1. Explore how to combine these concepts
2. Design a new challenging problem that integrates them

you must choose one of the following variation strategies:
```

```
1. Sequential Combination: Concepts are loosely chained or simple spliced.
2. Deep Integration (Fusion): Concepts are merged in a non-trivial, deeply interconnected
      way.

Output format(strictly follow):
 ## Part 1: Original Problems and Solution Analysis
 Step1: [Describe the steps of reasoning]
 Step2: xxx
 ...

 ## Part 2: New Problem Description:
 New_problem: [Describe the new problem clearly in natural language.]

 Input Format: [Specify the input format]
 Output Format: [Specify the output format]

 ## Part 3: Example Test Cases
 Input: [Input for test case 1]
 Output: [Expected output for test case 1]
 Input: [Input for test case 2]
 Output: [Expected output for test case 2]

 ## Part 4: Category
 difficulty: [Easy/Medium/Hard]
 tags: [tags of new problem, separated by commas.]
 variation: [cross-type or same-type fusion; Sequential Combination or Deep Integration.]

Note:
1. Please generate a difficult and original question.
2. The new problem must be rigorous and clearly stated, and include explicit input/output
      specifications or constraints.
3. Please design questions that have one correct answer; avoid 'output one possible
     combination' that could have multiple valid answers.
4. Provide two example test cases to demonstrate the new problem.
```

*Listing 5.* Prompts for Atomic Variants

```
You are an expert competitive programmer.
I'll provide you with one programming problem, its solution, and the key concepts they
    test.
You need to:
1. Analyze its problem design approaches
2. Create a new variation question based on the original one, You must choose **one** of
    the following variation strategies:
   1. **Increase Data Scale:** Significantly increase the constraints, ensuring that an
       optimized algorithm can still solve the problem within a 5-second time limit.
   2. **Rule Transformation (Preserving Core Algorithm):** Change the rules or
       constraints without affecting the fundamental algorithmic approach (e.g., in a "
       Climbing Stairs" problem, change steps from {1, 2} to {2, 3}).

Output format(strictly follow):
 ## Part 1: Original Problems and Solution Analysis
```

```
Step1: [Describe the steps of reasoning]
Step2: xxx
...

## Part 2: New Problem Description:
New_problem: [Describe the new problem clearly in natural language.]

Input Format: [Specify the input format]
Output Format: [Specify the output format]

## Part 3: Example Test Cases
Input: [Input for test case 1]
Output: [Expected output for test case 1]
Input: [Input for test case 2]
Output: [Expected output for test case 2]

## Part 4: Category
difficulty: [Easy/Medium/Hard]
tags: [tags of new problem, separated by commas, referring to the tags of the original
    problems.]
variation: [The type of variation used: Increase Data Scale or Rule Transformation]

Note:
1. The new problem must be rigorous and clearly stated, and include explicit input/output
    specifications or constraints.
2. Provide two example test cases to demonstrate the new problem.
3. Select the variation type you are most confident in, aiming to use each of the two
   types with roughly equal probability over multiple interactions.
4. Please design questions that have one correct answer; avoid 'output one possible
   combination' that could have multiple valid answers.
```

## B.9. Integrated Prompts for Error Analysis

*Listing 6.* Prompts for Error Analysis

```
Task: You are an expert coding problem analyst. You will be given Original Problems, a
   New Variant Problem, Correct Code, and Wrong Code from different models. Follow the
   steps below.

---

#### Step 1: Problem Analysis
*   Input: I will provide 1-2 original problems and 1 new variant problem.
*   Action: Analyze the knowledge points and key testing points for both the original and
     new problems.
*   Output Format:
    > Original Problem 1: [Title]
    > - Key Test Point: [Description]
    > - Knowledge Point: e.g., Segment Tree, Dynamic Programming
    >
    > Original Problem 2: [Title] (if provided)
    > - Key Test Point: [Description]
    > - Knowledge Point: [Description]
    >
```

```
    > New Variant Problem: [Title]
    > - Key Test Point: [Description]
    > - Knowledge Point: [Description]

---

#### Step 2: Code Error Analysis
*   Input: I will provide one correct code solution and several wrong code solutions from
     specific AI models.
*   Action: For each wrong code, analyze its errors by selecting from the predefined
    Error Taxonomy below. You must identify all errors that the model made.
*   Error Taxonomy (Choose from these 4 types):
    1.  Modeling Error: Selecting an incorrect or unnecessary algorithmic paradigm due to
         misinterpreting the problem variant (e.g., using greedy instead of DP).
    2.  Logic/Merge Bug: Wrong conditional logic or incorrect variable usage when merging
         subproblem results.
    3.  Indexing/Caching Bug: Incorrect cache sizing, array bounds, or off-by-one errors
         leading to crashes or incorrect outputs.
    4.  Complexity Error: Using an algorithm with unnecessarily high time/space
         complexity (e.g., $O(n^2)$ where O(n) suffices).
    5.  Others: Content is not code, or other implementation errors.
*   Output Format for Each Wrong Model:
    > Incorrect model: [model name]
    > - Error Type: e.g, Modeling Error
    > - Reason: [Concise explanation]
    > - Erroneous Code Snippet: Code
    > Incorrect model: [model name]
    > - Error Type: e.g., Logic/Merge Bug, Complexity Error
    > - Reason: [Concise explanation]
    > - Erroneous Code Snippet: Code

---

Tips:
Error Types can only be selected from Modeling Error, Logic/Merge Bug, Indexing/Caching
    Bug, Complexity Error, Others.
```

