# OpenReview forum: "UniCode: Augmenting Evaluation for Code Reasoning"
_ICML.cc/2026/Conference — ICML 2026 regular_

### Official Review · Reviewer_pRbA · 2026-03-08

**Soundness:** 3
**Presentation:** 3
**Significance:** 3
**Originality:** 3
**Overall Recommendation:** 4
**Confidence:** 4

**Summary:**

This paper introduces a generative evaluation framework to systematically test code reasoning limit of LLM. Unicode address "evaluation paradox", where models achieve high scores on traditional benchmarks but fail in real-world scenarios due to data contamination and other reasons. UniCode applies multi-dimensional augmentations to transform seed problems into complex variations. To evaluation such problems, Unicode develop a stress-driven synthesis pipeline that relies on brute-force solvers, majority voting, and LLM adjudication to generate reliable test cases without human curation. Through experiments, the authors demonstrate an average performance collapse of over 30% on the augmented problems.

**Compliance With Llm Reviewing Policy:**

Affirmed.

**Final Justification:**

The authors' rebuttal successfully addressed my primary concerns regarding the pipeline's yield and correlated failures. Because of that, I will raise my score.

**Key Questions For Authors:**

1. How do you guarantee that the reference solutions used to set the time and memory achieve the theoretically optimal time complexity? If a problem strictly requires $O(n \log n)$ but the candidate pool $\mathcal{O}_{valid}$ only contains $O(n^2)$ solutions, how does the pipeline prevent setting an excessively loose TL?
2. Have you conducted any human annotations to verify the accuracy of the LLM-generated error categorizations in Figure 4? If so, what is the agreement rate between the LLM-as-Judge and the human? The authors only mention a "98.2% validity rate" (Section 4 and Appendix B.4) for generated benchmark.
3. Among your generated benchmark, what is the exact ratio of your easy, medium and hard questions?

**Limitations:**

There are two major limitations:
1.There is a fundamental gap in its premise: although the authors aim to evaluate the open-ended nature of "real-world software engineering", the benchmark is exclusively derived from competitive programming platforms like Codeforces and LeetCode.
2. The ground-truth generation pipeline relies on a strict majority vote among LLM solvers for large-scale inputs. This creates a paradox for hard problems: since many models exhibit near-zero accuracy on these complex tasks, are they capable enough to form reliable and correct consensus.

**Strengths And Weaknesses:**

Strength:
1. Doing beyond shallow level adversarial perturbation (such as renaming the variables) to deeper algorithmic structural shift (like concept fusion). It's an evolution for current LLM code benchmarks
2. Strong theoretical and statistical foundation (author mentioned in Appendix B.5) for estimating the total error bound in the presence of erroneous tasks, giving credibility to automated test pipelines.
3. The deep analysis on "seed-problem regression" demonstrates that the models rely heavily on near-neighbor rather genuine logical synthesis

Weakness:
1. Section 4.1 states that 25000 seed problems were curated, yet the benchmark ultimately evaluates models on only 492 candidate problems (only 2% of questions are selected). If the pipeline discards problems simply because the generator fail to produce a valid brute-force solver or cannot reach consensus on large inputs, the benchmark suffers from severe survivorship bias. It artificially caps the benchmark's difficulty to the ceiling of the generator models, directly contradicting the claim of an "effectively infinite" problem space.
2. The automated diagnostics are unvalidated. In section 5.2, it mentioned a fine-grained error taxonomy, which is fully determined LLM prompting. There is no human baseline to validate this LLM-as-Judge metric.
3. For stage 2 of the ground-truth construction, the voting assumes the agreement implies correctness. However, as the paper mentioned, since LLMs suffer from "seed-problem problem", if multiple LLM solvers default to the same flawed seed logic, they will reach a consensus on incorrect outputs. The paper does not provide analysis on this correlated failure rate. In addition, since the vast majority of LLMs simply cannot write correct code for hard problems (in Table 1, even o4-mini could only answer 21.6% of hard questions), they will absolutely be unable to pool enough correct answers to reach a "majority consensus." This means that while this automated pipeline runs smoothly when generating easy and medium problems, it will highly likely get stuck on the hard level problems where breakthroughs are truly needed
4. Over-reliance on LLM generator.

---

> ### Author Rebuttal · Authors · 2026-03-29
>
> We thank the reviewers for their constructive feedback and for recognizing UniCode’s "deep algorithmic structural shifts" and "strong theoretical foundation." We note that several concerns center on the quality of the generated benchmarks and provide a general response below:
>
> 1. **High Yield**: We achieve a 94.8% generation success rate (not 2%), preserving the vast majority of challenging tasks.
> 2. **Mitigating Correlated Failures**: We use brute-force solvers to check answers. This filter drops the correlated error rate from 18.7% down to 5.2%.
> 3. **Human Alignment**: Our framework incorporates human verification at every stage, further supported by theoretical foundations for error bounds.
>
> ---
>
> > Q1: Does the 2% yield (492/25,000) due to generator failure introduce survivorship bias and cap difficulty at the generator's ceiling?
>
> We clarify that the 492 problems are not the maximum yield of our pipeline, but a high-quality curated subset. Our pipeline's actual generation rate is 94.8% (problems with validated suites in Table 3). The 25,000 seeds represent the **total space** to ensure diversity, and we sampled **~600 instances across 15 tags** and applied rigorous filtering to reach final 492. Our method generates complex variations, as SOTA reasoning LLMs achieve only a **48.2% average Pass@1**.
>
> > Q2 & Key.2: What is the agreement rate between the LLM-as-judge and the human for LLM-generated error categorizations in Figure 4?
>
> On 30 sampled failed tasks, the **LLM-human alignment is 93.4%**, with 94.5% inter-human agreement among 5 experts. We will include these statistics in the final version.
>
> > Q3 & Lim.2: If multiple LLM solvers share flawed seed regression (e.g., o4-mini’s 21.6% accuracy on hard problems), they may reach a consensus on incorrect outputs. What is the correlated failure rate?
>
> We mitigate correlated failures by using **Brute-Force (BF) solvers as an oracle**. BF solvers, using exhaustive search, naturally avoid the logical modeling errors common in "seed regression." By filtering optimized solvers against BF results, the correlated failure rate is reduced from 18.7% to 5.2% on hard problems. To further ensure "gold" quality for the most challenging tasks, we **manually verified all 115 "Extremely Hard" cases**, identifying and removing 9 invalid ones. This combined "BF-filtering + human cleaning" ensures a reliable consensus even when models exhibit low zero-shot accuracy.
>
> > Q4: Over-reliance on LLM generator.
>
> UniCode utilizes the LLM generator, a widely-adopted paradigm in recent benchmarks (e.g., MT-bench [ACL'24], MCU [ICML'25 Spotlight])—but significantly enhances it with a **physical verification layer**. Reliability is grounded in: (1) **Human Alignment**: problem quality (App. B.4), test case quality (App. B.1), and error classification (Q2) all demonstrate high congruence with humans; (2) **External Validity**: strong correlation ($r=0.986$) with LiveCodeBench; and (3) **Statistical Rigor: mathematically derived error bounds provide theoretical validity** (App. B.5).
>
> > Key Q1: If a problem strictly requires $O(n \log n)$ but the candidate pool $\mathcal{O}_{valid}$ only contains $O(n^2)$ solutions, how does the pipeline prevent setting an excessively loose TL?
>
> We guarantee optimal time complexity by utilizing **large-scale boundary inputs** and **runtime heuristics** (e.g., 5s) to prune suboptimal algorithms. As shown in our case study (App. B.6), if an $O(n^2)$ algorithm is used where $O(n \log n)$ is required, large edge cases cause execution time to surge (e.g., from 5s to 50s in Table 7), triggering the filter. Furthermore, our statistical analysis reveals that for **96.2% of 80 sampled problems, at least one candidate achieved the theoretically optimal complexity**.
>
> > Key Q3: Among your generated benchmark, what is the exact ratio of your easy, medium, and hard questions?
>
> Easy: 32%, Medium: 39%, Hard: 29%.
>
> > Lim.1: Although the authors aim to evaluate the open-ended nature of "real-world software engineering," the benchmark is exclusively derived from competitive programming platforms like Codeforces and LeetCode.
>
> We thank the reviewer for this insightful comment. Our intent is to use scalable augmentation methods to generate structurally novel tasks that simulate open-ended requirement evolution. Concepts such as "Concept Fusion" and "Efficiency Scaling" are designed to mirror the system scalability challenges and requirement shifts typical in real-world engineering. We agree that the term "real-world software engineering" was overly broad and **will refine it to "complex algorithmic reasoning in evolving scenarios"** to more precisely reflect the benchmark's focus.
>
> ---
>
> We sincerely thank the reviewers for these comments, which have significantly strengthened the rigor of our work. Please let us know if you have any further questions; we are happy to engage in further discussion to clarify any remaining points.

---

> > ### Author Rebuttal · Reviewer_pRbA · 2026-04-03
> >
> > The authors provided a comprehensive rebuttal and fully answer my concerns. I will raise the score. But please ensure that you explicitly acknowledge the manual curation of the extremely hard cases and the expanded validation sample size in the camera-ready version.

---

> > > ### Author Response · Authors · 2026-04-03
> > >
> > > We sincerely thank the reviewer for recognizing our work and for the score increase. We will incorporate the manual curation of hard cases and expanded validation samples in the final camera-ready version.

---

### Official Review · Reviewer_k7hd · 2026-03-09

**Soundness:** 3
**Presentation:** 4
**Significance:** 3
**Originality:** 3
**Overall Recommendation:** 5
**Confidence:** 4

**Summary:**

The authors introduce UniCode, an evaluation framework that allows systematic perturbation of coding benchmarks via multiple dimensions. Unlike traditional benchmarks, which are either static or rely on surface-level perturbations, UniCode provides a method to perturb questions at the algorithm level in a generative manner. Empirical evidence suggests that UniCode is challenging for state-of-the-art Large Language Models and Large Reasoning Models.

**Compliance With Llm Reviewing Policy:**

Affirmed.

**Final Justification:**

My concerns have been adequately addressed.

**Key Questions For Authors:**

I don't have additional questions.

**Limitations:**

yes

**Strengths And Weaknesses:**

Strengths:

1. Soundness: The paper is technologically sound. The authors construct systematic perturbations and verify their qualities by both LLMs and human expert annotators. The experiment is also rigorously designed to show model performance drops in different task complexities and task genres.

2. Presentation: The presentation is clear and easy to follow. The authors nicely compiled results in different figures and tables, which contain rich information in a readable manner.

3. Significance: The investigated question is interesting and important. Coding is one of the most important application areas for LLMs, and it is vital to ensure that the models perform reliably across different variations of known algorithms.

4. Originality: The proposed method is novel and verified via multiple methods.

Weaknesses:

1. Soundness: The paper seems to lack explicit arguments about statistical significance. I personally feel it is important to present statistical significance tests in perturbation papers when they talk about performance deltas.

2. If I understand it correctly, the benchmark primarily focuses on code generation. Do the authors think the proposed method can also be applied in code comprehension and other related tasks?

---

> ### Author Rebuttal · Authors · 2026-03-29
>
> We sincerely thank the reviewer for their positive assessment and for recognizing the soundness, originality, and significance of UniCode. We are encouraged that the reviewer found our methodology technically sound and the investigated question vital for the LLM community.
>
> ---
>
> >Q1. It is important to present **statistical significance tests** in perturbation papers when they talk about performance deltas.
>
> We appreciate this constructive suggestion. To address this, we conducted paired t-tests on a representative sample of 132 problems across three state-of-the-art models (o4-mini, Gemini 2.5 Flash, and GPT-4.1-mini).
>
> The performance drops from seed problems to UniCode-augmented variants are highly significant, with p-values of $2.8 \times 10^{-4}$, $3.2 \times 10^{-5}$, and $5.8 \times 10^{-5}$ respectively (**all $p < 0.05$**). This statistically confirms that the observed degradation reflects **inherent reasoning flaws** rather than random variance. We will include a dedicated section titled "Statistical Significance Analysis" in the revised manuscript.
>
> ---
>
> >2. Do the authors think the proposed method can also be applied in **code comprehension** and other related tasks?
>
> This is an insightful observation. While our current focus is code generation, the UniCode framework is **task-agnostic**, as the multi-dimensional augmentation operates at the algorithmic level.
>
> We believe UniCode is highly applicable to code comprehension. By presenting a model with an original algorithm alongside its augmented variant (e.g., with modified pruning or efficiency constraints), we can task the model with identifying the specific logic shifts. This ensures the model understands the underlying logic rather than merely recognizing patterns. Crucially, since these variants are generated programmatically using predefined rules, the **ground-truth labels** for such tasks are **obtained automatically**.
>
> Beyond comprehension, UniCode holds significant potential for Automated Program Repair (APR) and Vulnerability Detection. It can be used to synthesize subtle, logic-level "near-miss" bugs to stress-test the robustness of bug-finding tools. We will add a section in the revised manuscript to discuss these broader software engineering applications.
>
> ---
>
> Thank you again for your constructive feedback. Please let us know if you have any further questions.

---

> > ### Author Rebuttal · Reviewer_k7hd · 2026-04-01
> >
> > My concerns have been adequately addressed.

---

> > > ### Author Response · Authors · 2026-04-03
> > >
> > > Thank you for your recognition of our work. Thanks again for your time and effort in reviewing our paper!

---

### Official Review · Reviewer_Gy4e · 2026-03-12

**Soundness:** 2
**Presentation:** 3
**Significance:** 2
**Originality:** 2
**Overall Recommendation:** 3
**Confidence:** 4

**Summary:**

The paper introduces UniCode a framework designed to augment the evaluation of code generation models. It utilizes an automated pipeline to generate new programming tasks using existing seed problems aiming to provide a dynamic benchmark that mitigates data contamination. The methodology involves task generation code synthesis and correctness adjudication.

**Compliance With Llm Reviewing Policy:**

Affirmed.

**Key Questions For Authors:**

1. How does the framework explicitly prove that models are not simply recognizing and exploiting memorized structural patterns from the TACO seed problems?
2. Why was a comprehensive empirical comparison with Evol-Instruct omitted and how does UniCode theoretically and practically differ from it?
3. Can the authors demonstrate the robustness of the benchmark by substituting o4-mini with open-weights models for task generation and adjudication to eliminate closed-source bias?
4. What specific steps will be taken to broaden the quality assurance mechanisms beyond the current narrow scope to ensure comprehensive edge-case validation?

**Limitations:**

Yes.

**Strengths And Weaknesses:**

Strengths:
1. The framework proposes an automated pipeline for generating dynamic programming benchmarks which is a relevant approach to addressing the static nature of current evaluation datasets.
2. The focus on evaluating code generation models reflects an important and active area of research.

Weaknesses:
1. The proposed benchmark fundamentally fails to resolve the pervasive issue of training data memorization. Because the generated problems are derivative of existing distributions language models can still rely on memorized structural patterns rather than demonstrating genuine zero-shot reasoning.
2. The methodology lacks a critical comparison with Evol-Instruct. Since Evol-Instruct is a prominent established baseline for generating complex coding instructions the absence of this comparison significantly diminishes the empirical validation of the proposed augmentation technique.
3. The entire evaluation and generation pipeline relies excessively on a single closed-source model specifically o4-mini for both task creation and final adjudication. This introduces a severe model-specific bias and raises concerns about the reproducibility and robustness of the benchmark.
4. The framework demonstrates a heavy dependence on seed problems extracted from the TACO dataset. This reliance constrains the diversity of the generated tasks and limits the generalizability of the benchmark to other programming paradigms or difficulty levels.
5. The experimental section features highly limited ablations. The isolated impact of the various pipeline components remains unclear making it difficult to assess which mechanisms actually contribute to the claimed improvements.
6. Quality assurance within the framework remains notably narrow. The reliance on basic test cases or model-based verification does not guarantee the rigorous semantic correctness or edge-case handling required for a robust code evaluation benchmark.

---

> ### Author Rebuttal · Authors · 2026-03-29
>
> We sincerely thank the reviewer for the thoughtful feedback. We would like to clarify that many of the raised concerns are already addressed in our current submission. We provide detailed responses below:
>
> >Q1 & W1: How does the framework prove models are not simply exploiting memorized structural patterns from seed problems?
>
> Our extensive results demonstrate that UniCode effectively disrupts fixed structural patterns, requiring genuine zero-shot reasoning:
> - **Performance Collapse**: Models suffer a $>30\%$ performance drop across five dimensions on UniCode tasks (Fig. 1(e)). This significant degradation confirms that models cannot rely on memorized patterns from original distributions.
> - **Regression Phenomenon**: In Section 5, we conduct a deep analysis of code reasoning and identify the "Seed-problem Regression Phenomenon." We prove that UniCode can identify whether LLMs are performing genuine reasoning on new logic structures or merely recalling memorized seeds (Fig. 5).
> - **Error Analysis**: Our fine-grained analysis reveals that model failures on augmented tasks are predominantly "Modeling Errors" rather than simple implementation slips. This indicates a fundamental challenge in understanding the transformed task structures.
>
> >Q2 & W2: The methodology lacks a critical comparison with Evol-Instruct.
>
> We have cited and discussed Evol-Instruct in the Related Work (Lines 438-419): "Unlike traditional instruction-tuning or Self-Instruct (e.g., Xu et al., 2024; Luo et al., 2023), which focus on surface-level perturbations or linear extension of reasoning steps, UniCode targets underlying algorithmic transformations to disrupt memorized logic and evaluate true generalization.”
> Specifically, **Evol-Instruct focuses on data synthesis via linguistic complexity, while UniCode probes the depth of structural reasoning** via systematic logical transformations.
>
> >Q3 & W3: Can the authors demonstrate the robustness of the benchmark with open-weights models for generation and adjudication to eliminate closed-source bias?
>
> We have conducted experiments using the open-weight model DeepSeek-R1 for both generation and adjudication (App. B.3 Analysis of Generator Bias). We found that the source model does not introduce significant bias; model rankings remain highly consistent, with a **Pearson correlation of $r=0.984$** regardless of the generator used.
>
> >W4: The framework’s heavy reliance on TACO seed problems limits task diversity and the benchmark's generalizability across different paradigms and difficulty.
>
> We would like to clarify that our seeds are not solely dependent on TACO. As stated in Line 259, seeds are curated from a diverse range of platforms (including LeetCode and CodeForces). These seeds cover **9 top-level tags, 31 subtags, and 161 skills** (App. A.6), providing the necessary diversity to ensure generalizability across paradigms and difficulty levels.
>
> >W5: Impact of Pipeline Components (Ablation Study).
>
> We have provided a comprehensive ablation study in App. B.1 (Test Case Quality and Ablation). This section isolates the contribution of each mechanism:
> - Stage 1 (Brute-Force): We find that brute-force validation on small inputs is pivotal for correctness, yielding a **+7.1% absolute improvement** (from 86.7% to 93.8%) by filtering erroneous solver candidates (Lines 1138-1140).
> - Stages 2 & 3: Majority Voting (Stage 2) and LLM Adjudication (Stage 3) further refine the test suite, progressively enhancing both accuracy and edge-case coverage to ensure the overall robustness of the benchmark.
>
> >Q4 & W6: Quality assurance is narrow. Relying on basic test cases or model-based verification fails to ensure the semantic rigor and edge-case handling essential for a robust code benchmark.
>
> UniCode does not rely on "basic" test cases; instead, we implement a rigorous quality control mechanism:
> 1. **Tiered Test Generation**: As detailed in Sec 3.1, we specifically architect $G_{corn}$ (Corner Cases) and $G_{adv}$ (Adversarial Cases) to target boundary conditions that random tests ($G_{rand}$) often bypass.
> 2. **Rigorous Quality Validation**: Validation against human-curated data confirms 94.5% accuracy and 86.0% coverage, with theoretical error bounds provided (App. B.1 & B.5).
> 3. **Quantifiable Impact**: Table 1 reports the "$\Delta$ test impact," quantifying how our **$G_{adv}$ (avg. -7.03%) and $G_{corn}$ (avg. -7.35%)** significantly expose the model's sensitivity.
>
> We hope these clarifications, which highlight key sections in our submission (e.g., Sec 5, App. B), address the reviewer's concerns. We respectfully invite the reviewer to reconsider the merits and technical depth of our work.

---

> > ### Author Rebuttal · Reviewer_Gy4e · 2026-04-04
> >
> > I appreciate the authors' response. However, I still find the claim that 'UniCode effectively disrupts fixed structural patterns, requiring genuine zero-shot reasoning' to be too strong. The model failures on augmented tasks are predominantly 'Modeling Errors,' which may well be influenced by memorization. Consequently, models less susceptible to memorization might actually achieve higher scores, suggesting that comparisons on newer seed tests would be more meaningful. Furthermore, the fact that o3-mini outperforms DeepSeek-R1 in the first and second columns of Table 5 may precisely highlight a bias issue, given that o3-mini and o4-mini are likely trained on overlapping data sources. I am inclined to raise the soundness score to 3.

---

> > > ### Author Response · Authors · 2026-04-04
> > >
> > > We thank the reviewer for increasing the soundness score. We address the remaining concerns below:
> > >
> > > > 1: The claim is too strong.
> > >
> > > **We did not make this claim in our paper.** In our effort to provide a point-to-point response during the previous discussion phase, we may have misapplied the reviewer’s terminology. We now refine the wording to reflect the actual claim of our paper: UniCode disrupts fixed algorithmic patterns—ranging from surface-level shifts to structural reasoning graph reconfigurations—to expose model fragility and foster reasoning-oriented code intelligence.
> > >
> > > > 2: UniCode may not disrupt patterns; LLMs may well be influenced by memorization (as evidenced by "modeling errors").
> > >
> > > We argue that "Modeling Errors" (e.g., misapplying an algorithmic paradigm) are **empirical proof** of UniCode's **successful structural disruption**. These errors suggest that models are effectively "trapped" in the reasoning paradigms of the seed problems, exhibiting a seed-problem regression phenomenon. This confirms that our framework has successfully altered the reasoning graph topology, ensuring that **models cannot rely on statistical shortcut to reach a correct solution**.
> > >
> > > > 3：Comparisons on **newer seed tests** would be more meaningful.
> > >
> > > The use of old problems as seeds is a **deliberate experimental control**. To confirm that a performance drop on a variant ($S'$) stems from a reasoning failure, we must first establish that the model "knows" the underlying concept via high performance on the seed ($S$). Using entirely new seeds would **make it impossible to distinguish between a lack of domain knowledge and a failure of logical generalization**. However, as an evolving benchmark, UniCode is designed to continuously integrate newly released problems to maintain a diverse seed bank.
> > >
> > > > 4: o3-mini outperforms DeepSeek-R1 in Table 5 and may highlight a bias issue.
> > >
> > > We apologize for a typo in the original text: o3-mini’s performance under the o4mini generator is 55.1% (not 57.1%); we will correct this in our final version. Crucially, **model rankings remain identical** across all generators, which empirically refutes significant generator bias. Furthermore, we conducted a "Multi-LLM Mixed" validation (using R1, o4-mini, GPT-5, and Gemini-2.5) on 90 problems. This ensemble approach effectively **smooths individual model biases and stabilizes task difficulty**, as shown below:
> > >
> > > | Model (Evaluated) | UniCode (Multi-LLM Mixed) | UniCode (DeepSeek-gen) | UniCode (o4mini-gen) | LiveCodeBench (Human) |
> > > | :--- | :---: | :---: | :---: | :---: |
> > > | gpt-5 | 68.80% | 72.50% | 67.70% | — |
> > > | o4-mini | 67.70% | 70.20% | 66.90% | 74.20% |
> > > | deepseek-r1 | 60.00% | 61.60% | 56.60% | 73.10% |
> > > | o3-mini | 55.60% | 51.00% | 55.10% | 63.00% |
> > > | gpt-4.1-mini | 44.50% | 41.30% | 42.40% | 53.20% |
> > >
> > > We appreciate the reviewer's time and effort, and we hope our response addresses your concerns.

---

### Official Review · Reviewer_KnpB · 2026-03-16

**Soundness:** 2
**Presentation:** 3
**Significance:** 3
**Originality:** 2
**Overall Recommendation:** 3
**Confidence:** 4

**Summary:**

This paper presents UniCode, a generative evaluation framework designed to assess the code generation abilities of LLMs. Starting from competitive programming problems as seed tasks, the framework systematically generates new problem variants along five augmentation axes: narrative perturbation, rule modification, efficiency scaling, sequential composition, and concept fusion. To obtain reliable ground-truth outputs without relying on human-written solutions, the authors build a stress-driven test generation pipeline that combines brute-force filtering, consensus validation, and LLM-based adjudication. Using UniCode, they evaluate 19 LLMs and observe a substantial performance drop—an average decline of 31.2% when models move from the original seed problems to the augmented variants.

**Compliance With Llm Reviewing Policy:**

Affirmed.

**Final Justification:**

Please refer to my response to the Area Chair's comment above. I maintain my original weak reject recommendation.

**Key Questions For Authors:**

1. How does UniCode compare to simply collecting new problems over time (LiveCodeBench approach)? Specifically, on the overlap models between UniCode and LiveCodeBench, does UniCode's augmentation-based approach reveal failure modes that LiveCodeBench misses, or vice versa?

**Limitations:**

yes

**Strengths And Weaknesses:**

### Strengths

1. Separating augmentations into five axes allows attributing failures to specific reasoning dimensions rather than reporting a single aggregate score. The empirical result that narrative perturbation causes minimal degradation while sequential composition and concept fusion cause >30% drops (Figure 3) is a concrete, reproducible finding.

2. 19 models spanning reasoning/non-reasoning, open/closed-source, and varying parameter scales. The correlation analysis with LiveCodeBench (r=0.986) and LiveCodeBenchPro (r=-0.916) in App. A.4 provides external validity.

### Weaknesses

1. All five augmentation types — rewriting the story, tweaking constraints, scaling input size, chaining subproblems, and merging concepts — are standard techniques that any experienced competitive programming problem setter employs routinely. The paper's contribution is automating this via LLM prompts (Listings 4–5, App. B.8), but the prompts themselves are straightforward ("analyze their problem design approaches" / "create a new challenging problem"). No new algorithmic or theoretical insight is introduced for controlling augmentation quality or difficulty.

2. The entire pipeline — problem generation, test case generation, ground-truth construction, and error classification — relies on LLMs. This creates a fundamental tension: the framework claims to rigorously evaluate LLM reasoning, but its own reliability depends on LLM correctness. The 5.5% test case error rate and 14% missed-coverage rate are non-trivial. While the B.5 analysis bounds the impact of erroneous tasks theoretically, it assumes errors are randomly distributed; in practice, LLM-generated errors likely cluster on exactly the hard compositional variants where evaluation is most critical.

3. The paper generates 492 candidate problems from 25,000 seeds (Section 4.1) and then excludes "low-difficulty problems that were solved perfectly by all baseline models," but never states the final benchmark size.

4. Table 5 shows that deepseek-r1 scores 61.6% on its own generated problems vs. 55.6% on o4-mini-generated ones — a 6% gap suggesting stylistic self-preference. The proposed mitigation is mentioned but not evaluated. Since the main benchmark uses o4-mini as the primary generator, the reported scores may systematically underestimate o4-mini's relative difficulty and overestimate other models' weaknesses.

---

> ### Author Rebuttal · Authors · 2026-03-29
>
> We thank reviewers for recognizing UniCode's validity and findings. We address your concerns below.
>
> ---
>
> > **Key Question: Comparison with LiveCodeBench (LCB)**
>
> UniCode and LCB evaluate orthogonal dimensions: LCB targets data contamination, while UniCode probes structural reasoning depth via systematic logical transformations.
>
> 1. **Structural Dependency:** UniCode reveals that failures are not random; models frequently "regress" to seed-problem paradigms under logical variation (Fig 5)—a structural dependency that LCB **cannot detect** due to its lack of evolutionary problem ties.
> 2. **Diagnostic Granularity:** Unlike LCB’s binary scoring, UniCode provides fine-grained diagnostics showing that conceptual modeling errors are the primary drivers of performance collapse (Fig 4).
> 3. **Exploration of "Blind Spots":** Our generative pipeline explores reasoning combinations (e.g., Game Theory + Greedy) often overlooked by human designers, achieving superior coverage of the logical space (Fig 1(d)).
> 4. **Scalability:** UniCode offers a highly cost-effective ($0.041/prob) and scalable framework, making deep evaluation more accessible to the research community.
>
> ---
>
> > **Q1: Concern regarding the contribution.**
>
> We respectfully propose that the value of a benchmark can be grounded in three pillars: 1) addressing persistent evaluation challenges, 2) identifying critical dimensions, and 3) delivering unique diagnostic insights. UniCode achieves these via:
>
> 1. **Solving Evaluation Challenges:** Static benchmarks suffer from severe contamination and fixed paradigms. UniCode provides a unified framework to transform known algorithms into **complex variations at scale**, creating a dynamic, contamination-resistant benchmark.
> 2. **Identifying Critical Dimensions:** Our five axes target reasoning limits where SOTA LLMs remain fragile. Unlike surface-level perturbation (e.g., gsm-symbolic), UniCode systematically disrupts structural logic from novel perspectives to expose hidden vulnerabilities.
> 3. **Unique Diagnostic Insights:** UniCode detects **"seed-problem regression"** and fine-grained errors (e.g., modeling/logic bugs) that other code benches miss.
>
> Therefore, UniCode contributes a systematic framework featuring five augmentation dimensions and a reliable test generation pipeline, enabling scalable evaluation to expose LLM fragilities and provide rich diagnostic signals to guide future development.
>
> ---
>
> > **Q2: Reliability of LLM-in-the-loop Pipeline**
>
> Reliability stems from a **"Capability Gap"**: our generator (strongest models + brute-force + voting) consistently outperforms evaluated models. As established in recent literature (e.g., MT-bench [ACL'24], MCU [ICML'25-Spotlight]), LLM-involved pipelines are robust when they align with human judgment and preserve ranking consistency.
>
> 1. **Human-in-the-loop Validation:** At each step, we conduct human studies to verify problem quality (App. B.4), test case accuracy (App. B.1), and error classification (Fig 5), demonstrating that the output is highly congruent with expert.
> 2. **External Validity:** The **high correlation with LiveCodeBench (r=0.986)** and LiveCodeBenchPro (r=-0.916) in App. A.4 further confirms that our automated metrics reflect genuine model capabilities (as you mentioned in review).
>
> ---
>
> > **Q2: Test Case and Coverage Errors**
>
> UniCode’s **5.5% error rate** is competitive with gold-standard human benchmarks (e.g., SWE-bench $\approx$ 16.4%, ImageNet $\approx$ 6%), and can be further mitigated through the estimation methods described in App. B.5 (Eq. 4). Additionally, we performed a manual check of "Extremely Hard" (those where all LLMs failed) tasks and removed 9/115 invalid cases.
>
> ---
>
> > **Q3: Final Benchmark Size**
>
> The final benchmark comprises **492 tasks**, curated from ~600 problems by filtering trivial and problematic cases. Every task in the final release will undergo careful manual verification.
>
> ---
>
> > **Q4: Stylistic Self-preference and Bias Mitigation**
>
> Performance gains reflect shifts in task difficulty rather than "self-preference," as improvements were uniform across SOTA LLMs (gpt-5 +4.8%, o4-mini +3.3%, vs r1 +5.0%). Crucially, **model rankings remained consistent** across different generators. Further validation via "Multi-LLM Mixed" (r1, o4-mini, gpt-5, gemini-2.5) of 90 problems demonstrates its effectiveness in smoothing model bias and stabilizing task difficulty.
>
> | Model  | UniCode (Multi-LLM Mixed) | UniCode (DeepSeek-gen) | UniCode (o4mini-gen) | LiveCodeBench (Human) |
> | :--- | :---: | :---: | :---: | :---: |
> | gpt-5 | 68.80% | 72.50% | 67.70% | — |
> | o4-mini | 67.70% | 70.20% | 66.90% | 74.20% |
> | deepseek-r1 | 60.00% | 61.60% | 56.60% | 73.10% |
> | o3-mini | 55.60% | 51.00% | 57.10% | 63.00% |
> | gpt-4.1-mini | 44.50% | 41.30% | 42.40% | 53.20% |
>
> We thank the reviewers for their insights. We hope these clarifications address your concerns and would be happy to provide further information if needed.

---

> > ### Author Rebuttal · Reviewer_KnpB · 2026-04-03
> >
> > I thank the authors for the detailed rebuttal. The Multi-LLM Mixed experiment largely addresses the stylistic self-preference concern, and the benchmark size is now clear.
> > However, two core concerns remain unresolved:
> >
> > 1. Methodological novelty. The rebuttal reframes the contribution as a "systematic framework" with practical value, but does not address my original point that the augmentation axes and their LLM prompt implementations are standard competitive programming practices. The gap between a useful engineering artifact and a methodological contribution is not bridged.
> >
> > 2. Non-random error clustering. My concern was not about the aggregate 5.5% error rate, but whether errors systematically concentrate on the hardest compositional variants where evaluation matters most. The rebuttal compares to SWE-bench/ImageNet error rates and cites the App. B.5 bound, but that bound assumes random distribution — exactly the assumption I questioned.

---

> > > ### Author Response · Authors · 2026-04-03
> > >
> > > We thank the reviewer for the insightful follow-up. We are encouraged by your recognition of UniCode’s engineering value and that our "Multi-LLM Mixed" experiment resolved the stylistic preference concerns. Remaining points are addressed below.
> > >
> > > ---
> > >
> > > > **Q1: Methodological novelty.**
> > >
> > > Methodologically, UniCode is a search-based discovery framework that transforms traditional human design principles into a systematic exploration of high-dimensional reasoning spaces, supported by a stress-driven pipeline and human-in-the-loop verification.
> > >
> > > UniCode formalizes code reasoning evaluation as a **structured transformation space**. By utilizing the atomic and compositional axes as evolutionary transformation operators (see **Fig 1(a)**), it moves beyond discrete task collection toward a systematic mapping of reasoning boundaries to identify specific regions of model performance collapse.
> > >
> > > We contend that the combination of fundamental **evolutionary operators and a rigorous verification pipeline** holds significant potential for the field. Just like how **AlphaEvolve** utilizes search-based discovery to find novel scientific algorithms, UniCode leverages operator-controlled task evolution to uncover previously unknown model vulnerabilities and reasoning patterns.
> > >
> > > This reframing is essential because the value of an evaluation framework lies in the **new diagnostic signals** it reveals. To our knowledge, UniCode is the **first work to systematically reveal** the structural dependency between seed tasks and their variants—the “seed-problem regression” phenomenon, providing a methodology to explore compositional generalization far beyond standard human-designed distributions.
> > >
> > > ---
> > >
> > > > **Q2: Non-random error clustering.**
> > >
> > > We share the same concern. Therefore, we performed a manual check of the **“Extremely Hard”** tasks (those on which all LLMs failed) and **removed 9/115 invalid cases**, as demonstrated in our previous response. Furthermore, we conducted experiments by randomly sampling 60 instances per level, which confirms that error rates remain **well-bounded (<7%)** even as complexity increases.
> > >
> > > | Difficulty | Easy | Medium | Hard |
> > > | :--- | :--- | :--- | :--- |
> > > | **Error Rate** | 3.30% | 5.00% | 6.70% |
> > >
> > > ---
> > >
> > > We thank the reviewer for these insightful points, which have helped strengthen our work. We hope this response alleviates your concerns.

---

### Decision · Program_Chairs · 2026-04-30

**Decision:**

Accept (regular)

**Comment:**

We thank the authors for their submission, which all reviewers agree addresses an important problem in a systematic way. Measuring memorization and generalization of LLMs is a popular topic, and coding problems provide a useful environment in which to measure this because of the ease at which verifiable samples can be constructed.

Reviewers noted both in reviews and in discussions that there are a number of similar papers, leaving the novelty of the contribution here relatively low. But they mostly agreed that the practical contribution and reasonably comprehensive analysis, including a number of improvements offered during the rebuttal phase, warrants publication. Please carefully consider the remaining concerns, especially where the writing can be improved to better acknowledge and discuss limitations of this work.